# Climate variability in subarctic area for the last two millennia

Marie Nicolle[1], Maxime Debret[1,] Nicolas Massei[1], Christophe Colin[2], Anne deVernal[3], Dmitry Divine[4,5], Johannes P. Werner[6], Anne Hormes[7], Atte Korhola[8] and Hans W. Linderholm[9]

[1] Normandie Univ, UNIROUEN, UNICAEN, CNRS, M2C, 76000 Rouen, France

[2] GEOPS, CNRS, University of Paris-Sud, 91405 Orsay Cedex, France

[3] Centre de recherche en géochimie et géodynamique (Geotop), Université du Québec à Montréal, Montréal, QC, Canada

[4] Norwegian Polar Institute, Tromsø, Norway

[5] Department of Mathematics and Statistics, Arctic University of Norway, Tromsø, Norway

[6] Bjerknes Center for Climate Research and Department of Earth Science, University of Bergen, Bergen, Norway

[7] University of Gothenburg, Department of Earth Sciences, Gothenburg, Sweden

[8] Department of Environmental Sciences, Environmental Change Research Unit (ECRU), University of Helsinki, P.O. Box 65, 00014 Helsinki, Finland

[9] Regional Climate Group, Department of Earth Sciences, University of Gothenburg, 40530 Gothenburg, Sweden

*Correspondence to*: Marie Nicolle (marie.nicolle2@univ-rouen.fr)

**Abstract.** To put in perspective the recent climate change, it is necessary to extend the instrumental climate records with proxy data from palaeoclimate archives. Arctic climate variability for the last two millennia has been investigated using statistical and signal analyses from three regionally averaged records from the North Atlantic, Siberia and Alaska based on many types of proxy data archived in the Arctic 2k database. In the North Atlantic and Alaska areas, the major climatic trend is characterized by long-term cooling interrupted by the recent warming that started at the beginning of the 19th century. This

cooling trend is not clearly visible in the Siberian region. The cooling of the Little Ice Age (LIA) was identified from the individual series, but it is characterized by wide range spatial and temporal expression of climate variability, in contrary to the Medieval Climate Anomaly. The LIA started at the earliest by around 1200 AD and ended at the latest in the middle of the 20th century. The large spread temporal coverage of LIA did not show regional consistency or particular spatial distribution and did not show relationship with archive/proxy type either. A focus on the last two centuries shows a recent warming

characterized by a well-marked warming trend paralleling with increasing greenhouse gas emissions. It also shows a multi-decadal variability likely due to natural processes acting on the internal climate system at regional scale. A 16-30 years cycle is found in Alaska and seems to be linked to the Pacific Decadal Oscillation (PDO) whereas ~20-30 and ~50-90 years periodicities characterize the North Atlantic climate regime, likely in relation with the Atlantic Multidecadal Oscillation (AMO). These regional features are apparently linked to the sea-ice cover fluctuations through ice-temperature positive

feedback.

## 1. Introduction

Since the beginning of the industrial era, the global average temperature has increased by about 1°C and the recent decades have been the warmest in the last 1400 years (PAGES 2k Consortium, 2013; IPCC, 2013). The warming is more pronounced

at high latitudes of Northern Hemisphere than in other parts of the Earth (Serreze and Barry, 2011; PAGES 2k Consortium, 2013), being more than twice the rate and magnitude in the Arctic than the global average (Cohen et al., 2014). To replace this warming in the perspective of long-term natural climate variability, the instrumental time series are not sufficient and it is necessary to extend the meteorological measurements back in time with proxy data from in palaeoclimate archives (ice cores,

tree-rings, lake sediments, speleothems, marine sediments and historical series).

Over the last decade, extensive efforts were made to collect and compile palaeoclimate data available in order to reconstruct past climate variability at regional, hemispheric and global-scales. Most temperature reconstructions include different types of archives and proxies (Morberg et al., 2005; Mann et al., 2009; Kaufman et al., 2009; Ljungqvist, 2010; Marcott et al., 2013) and some studies focused on single palaeoclimate archive type and/or area (e.g. McGregor et al., 2015 for oceans; Weissbach

et al., 2016 for ice core; Wilson et al., 2016 for tree rings). In the Arctic and Subarctic area (90-60°N) several multi-proxy reconstructions of temperatures encompassing the last two millennia were published at global scale (PAGES 2k Consortium, 2013; McKay and Kaufman, 2014; Werner et al., 2017) and regional scale (Hanhijärvi et al., 2013). The annual resolution of these reconstructions allows the study of the climate variability from low frequencies (i.e. millennial and multi-centennial fluctuations) to high frequencies such as decadal variations.

Climatic reconstructions highlighted a millennial cooling trend associated to the monotonic reduction in summer insolation at high northern latitudes, and a reversal marked by an important warming of more than 1°C consistency with the increase of greenhouse gas since the mid-20th century (e.g. Kaufman et al., 2009; Pages 2k Consortium, 2013). The long-term cooling trend correlates with the millennial-scale summer insolation reduction at high northern latitudes (Kaufman et al., 2009) but an increased frequency of volcanic events during the last millennium may also have concur and contribute to the cooling episodes

that occurred after 1000 AD (PAGES 2k Consortium, 2013; Sigl et al., 2015).

Superimposed to the long-term climate fluctuation, continental-scale temperature reconstructions in the Northern Hemisphere highlight major climatic warming and cooling pulses during the last millennium, with relatively warm conditions during the Medieval Climate Anomaly (MCA, 950-1250 AD, Mann et al., 2009) and a cold Little Ice Age (LIA, 1400-1700 AD, Mann et al., 2009) period. The LIA is, however, characterized by an important spatial and temporal variability, particularly visible

at more regional scale (e.g. Pages 2k Consortium, 2013). It has been attributed to a combination of natural external forcings (solar activity and large volcanic eruptions) and internal sea-ice/ocean feedbacks which fostered long-standing effects of short-lived volcanic events (Miller et al., 2012).

Arctic-subarctic multidecadal climate variability is also influenced by internal climatic system dynamic such as the Atlantic Multidecadal Oscillation (AMO) or the Pacific Decadal Oscillation (PDO) which may impact temperatures and sea-ice cover

fluctuations (Chylek et al., 2009). The reconstruction of these oscillations with paleoclimate records offers the possibility to explore the linkages between the internal climate variability and the Arctic-subarctic climate over the last two millennia (e.g. Knudsen et al., 2014; Miles et al., 2014; Wei et Lohmann, 2012).

In this study, we explore the regional expression of the Arctic-subarctic climate variability during the last two millennia using statistical and wavelet analysis. To do so, we define three region, North Atlantic, Alaska and Siberia, from which we calculated

the climatic variations. Hence, the regional mean records allowed us to determine if the timing of the long-term and secular (MCA and LIA) climatic fluctuations which occur at the global Arctic-subarctic scale are also characteristic of the regional climate variability. A special attention is given to the last two centuries, with the comparison between the three regional mean records and instrumental climate index, to determine the influence of internal climate variability but also the ability of paleoclimate series to reproduce decadal to multidecadal variability observed in instrumental data.

## 2. Paleoclimate data

The records used in this study were compiled by the Arctic 2k working group of the Past Global Changes (PAGES) research programme. This working group released a database comprising 56 proxy records for the Arctic area (version 1.1.1, McKay and Kaufman, 2014). The database contains all available records that meet data quality criteria concerning location (from north of 60°N), time coverage (extend back to at least 1500 A.D.), mean resolution (better than 50 years), dating control (at least one age control point every 500 years) (Fig. 1a). See Table S1 in supplementary material for more informations about each site  to (cf. also McKay and Kaufman, 2014).

Proxy records are from different archive types. Most are continental archives with very reliable chronologies (16 ice cores, 13 tree rings, 19 lake sediment cores and 1 speleothem). Six records are from marine archives and one is a historic record (months of ice cover). Among the 56 records, 35 have an annual resolution (Fig. 1b). Hence, the high temporal resolution of the Arctic 2k database series offers the possibility to study the high frequency climate variability of the last two millennia, assuming that the proxy record climate variability and the archiving process do not induce a bias in the multi-annual to centennial frequencies analyzed.

The database has been built from palaeoclimate proxy series with demonstrated relationship to temperature variability. All the proxy data used have been published in a peer-reviewed journal and the sensitivity of each proxy record to temperature was evidenced either statistically (e.g. correlation with instrumental temperature data) or mechanistically with the description of the processes through which the proxy is shown its sensitiveness to temperature change (McKay and Kaufman, 2014).

Our review of the original publications presenting the data used to develop the Arctic 2k database led us to raise some concerns about the actual temperature controls on proxy. In some case, the correlation between proxy measurements and instrumental temperatures is significant but weak, with a correlation coefficient lower than 0.5 (e.g. Bird et al., 2009; D'Arrigo et al., 2005; D'Arrigo et al., 2009, Spielhagen et al., 2011; Wiles et al., 2014).  Such weak relationships suggest that the variability recorded by the proxies are not exclusively linked to the mean annual temperature but probably also relate to other parameters, climatic or not. In some cases, the authors clearly state that the relationship is not strong enough for reconstructing high resolution variations (D'Arrigo et al., 2005). As there are such uncertainties in the assumed temperature control on proxy, whatever the archive type, we choose to work on the original proxy records directly and not on temperature reconstructions derived from them.

## 3. Regional approach

The spatial distribution of the series highlights heterogeneity (Fig. 1a). among the 56 series of the Arctic 2k database, 40 (71%) are from the North Atlantic sector, including Scandinavia, Iceland, Greenland and Canadian Arctic, while 11 (20%) represent the Alaska region and 5 (9%) the Siberia. This high spatial discrepancy raises the question of the influence of the over weighting of the North Atlantic sector on the global Arctic signal. Therefore, in order to avoid a regional bias, we divide the Artic area into three sectors (Fig. 2a). The record n°9 located in the central part of the Canadian Arctic, between the North Atlantic and Alaska, was finally included to the North Atlantic regional mean record due to the correlation between ring width and Canadian-North American temperatures highlighted by the authors (D'Arrigo et al., 2009). The number of time series used for the Siberian regional averaged record is very low, with only 5 series for a large area and the statistical representativeness of the data is thus questionable.

Calculating regionally averaged records allows us to investigate the common spatial climate signal of each region and reduce the noise of individual records due to local effect (e.g. Weissbach et al., 2016). Before calculating each regional mean records, all records were standardized over the whole record to report the variations in terms of the standard deviation, which permits to compare the records with each other, regardless the parameters and unit values of independent records. The number of data points used to calculate each regional mean records is also indicated (Fig.2e-g). The three regional mean records based on the spatial distribution of the series were calculated and then compared with a global Arctic mean record presented at the figure 3. The correlation between the global Arctic record and the mean North Atlantic record shows particularly strong relationship (Figure 4b, $r^2 = 0.81$, p-value <<0.05). Correlations are weaker between the Arctic mean and the regional average Alaska record (Figure 4b, $r^2 = 0.23$, p-value <<0.05) or the regional mean Siberian record (Figure 4c, $r^2 = 0.16$, p -value <<0.05). The strong influence of the spatial distribution of data on the global mean Arctic record is also highlighted by the wavelet coherence analysis (see Appendix A for the method description). Wavelet coherence spectra revealed much stronger coherence between the North Atlantic sector and the global Arctic mean than for the two other regions, particularly at low-frequency with common variabilities for periods around 200 (170-220 years) and 500 years (395-540 years) which occur during all the last two millennia (Fig. 4d). The coherence spectra between global Arctic mean record and the Alaska regional record, shows a significant periodicity around 200 years only for an interval mostly spanning from 1330 to 1900 AD following (Fig. 4e). The coherence wavelet spectra between the global Arctic mean and the Siberian mean record does not highlighted significant periodicity around centennial scale except after 1680 AD (Fig. 4f). The comparison between regional mean records and the global Arctic subarctic record highlight climate variability dominated by North Atlantic signal, which is normal due to the much higher number of time series available in this area. For this reason, we decided to study the Arctic-subarctic climate variability for the last two millennia with a regional approach.

The grouping into the three regions (i.e. North Atlantic, Alaska and Siberia) is justified by present day regional climate. Theclimate of the Arctic- subarctic is influenced by the Atlantic and the Pacific oceans, which experience internal variability on different time-scales with specific regional climate impacts. In the North Atlantic sector, instrumental sea surface

temperature (SST) variations since 1860 AD highlight low-frequency oscillations known as the Atlantic Multidecadal Oscillation (AMO) (Kerr, 2000). The AMO corresponds to the alternation of warm and cool anomalies, which have considerable impact on the regional climate over the Atlantic, the North America and western Europe (e.g. Enfield et al., 2001; Sutton and Hodson, 2005; Knight et al., 2006; Assani et al., 2011). In the North Pacific, the Pacific Decadal Oscillation (PDO) drives the multidecadal variability (Mantua et al., 1997). It is defined as the leading principal component of monthly SST in the North Pacific Ocean (poleward of 20°N) (Mantua et al., 1997; Mantua and Hare, 2002). Positive phases of PDO are associated with precipitation deficit and positive temperature anomalies in the northwest United States (U.S.)and precipitation increases in southern Alaska and south western U.S. (Mantua and Hare 2002; Zhang and Delworth, 2015). Conditions are reversed during negative PDO phases.

## 4. Regional climate variability during the last two millennia

### 4.1. Long-term tendencies

Regional mean records for the three sectors and their corresponding 50-years LOESS filtering are presented in Figures 2 and 3. The North Atlantic and Alaska regional records show well-marked and significant decreasing trends before the beginning of the 19[th] century: $\tau$=-0.28 (p<<0.01; Fig. 5b) and $\tau$=-0.42 (p<<0.01; Fig. 5c), respectively. In the Siberian region, no decreasing trend is recorded ($\tau$=-0.02, p=0.20; Fig. 5d). These trends are also shown from the analysis of all individual records from that region (Fig. 5a and table S2). All the regions are characterized by significant warming after the beginning of the 19[th] century: $\tau$=0.40 (p<<0.01) for the North Atlantic, $\tau$=0.48 (p<<0.01) for the Alaska and $\tau$=0.45 (p<<0.01) for the Siberia.

The Subarctic North Atlantic regional record is characterized by two different trends. The first millennium does not show long-term fluctuations. However, it is marked by a cold event pulse at ~675 AD, which is depicted in the multi-proxy reconstruction of Hanhijärvi et al. (2013) from the Arctic Atlantic region and coincided with the occurrence of volcanic events (Sigl et al., 2015). The second millennium is characterized by a well-marked decreasing trend, particularly clear after ~1250 AD, and ending at ~1810 AD with the onset of the recent warming phase (Fig. 5b).

The first millennium in the Alaska region is characterized by a pronounced trend of decreasing temperatures until ~660 AD followed by an increase until the beginning of the second millennium (Fig. 5c). The cold minimum at ~ 660 AD is recorded in three time series over the five available. During the interval between ~1000 and ~1530 AD, temperatures decreased markedly, followed by a period of slight increase, before the recent warming starting at ~1840 AD in the Alaska area.

In contrary to the subarctic North Atlantic and the Alaska regional mean records, the Siberian regional mean record does not show apparent differences in trend between the first and the second millennium (Fig. 5d). The recent warming is well-marked in the Siberian area and started at ~1820 AD. Notable warm events occurred at ~250 AD, ~990 AD and ~1020 AD.

In the subarctic North Atlantic, the analysis of individual time series revealed inconsistency between the data from the marine record n°38, which is based on diatoms (Berner et al., 2011) and that of record n°39, which is based on alkenones (Calvo et al., 2002). The two data sets are from the same marine core (MD 95-2011) but suggest opposite trends before 1810 AD (Table

S2). Data from record n°38 shows a significant decreasing trend ($\tau$=-0.18, p<0.01) whereas those from record 39 presents a slightly increasing but non-significant trend ($\tau$=0.14, p=0.14). Different sensitivity to seasonal temperatures possibly explain difference between the two records as previously reported from the Nordic Seas (van Nieuwenhove et al., 2016). In the Arctic-subarctic areas, diatoms often relate to spring bloom whereas alkenones are produced by coccolithophorids which develop during the warmest part of the summer (e.g. Andruleit, 1997).

Except for some time series that record warming trends, which can be explained by local effects or differential seasonal responses, most individual series and regional mean records show decreasing trends before the beginning of the 19[th] century. The millennial-scale cooling trend is consistent with previously published reconstructions from North Atlantic (Hanhijärvi et al., 2013), Arctic (Kaufman et al., 2009; PAGES 2k Consortium, 2013; McKay and Kaufman, 2014) and the Northern Hemisphere (e.g. Morberg et al, 2005; Mann et al., 2008). A robust global cooling trend ending at about 1800 AD was also observed in regional paleoceanographic reconstructions (McGregor et al., 2015). The millennial cooling trend has been attributed to the reduction in summer insolation at high northern latitudes since the beginning of the Holocene (Kaufman et al., 2009), and associated to volcanic and solar forcings, notably during the last millennia (PAGES 2k Consortium, 2013; Stoffel et al., 2015). Whereas previous studies dates the transition between the long-term cooling and the recent warming at the beginning of the 20[th] century (e.g. Mann et al., 2008; PAGES 2k Consortium, 2013), we identified here that the cooling trend ended between 1810 and 1840 AD. The evidence of an industrial-era warming starting earlier at the beginning of 19[th] century was proposed by Abram et al. (2016) for the entire Arctic area. However, the intense volcanic activity of 19[th] century (1809, 1815, and around 1840, Sigl et al., 2015) may also explain the apparent early warming trend suggesting that it may have been recovery from a exceptionally cool phase.At the scale of the Holocene, internal fluctuations occurring at millennial scale have been identified in the subarctic North Atlantic area and were tentatively related to the ocean dynamics (Debret et al., 2007, Mjell et al., 2015). Therefore, to better understand the cooling trend of the last two millennia in a larger temporal context taking into account the role of oceanic variability on the long-term temperature variations, longer time series encompassing the entire Holocene would be useful.

## 4.2. Secular variability

Long-term change is not the only variability mode that defines the last two millennial climate, which was also characterized by long standing climatic events such as the Little Ice Age (LIA) and the Medieval Climate Anomaly (MCA). Here, we intent to summarize the expression of the LIA and the MCA in Arctic-subarctic area based on the Arctic 2k records. The timing of these two periods, which we identified in most series used in this study but not all of them, is taken from the original publications. The tables that list the beginning and ending of the LIA and the MCA are available in supplementary material.

The MCA corresponds to a relatively warm period occurring between 950 and 1250 AD (Mann et al., 2009). The starting year of this relatively warm period in the Arctic-subarctic area ranges between 900-950 AD in the Siberia and 900-1000 AD in Alaska, which is consistency with the overall records of the Northern Hemisphere (Fig. 6a). In the North Atlantic sector, the MCA began between 800 and 1050 AD, expected in two lake sediments records located in the Canadian Arctic in which the

MCA started at the end of the 12[th] century (Arc_25, Moore et al., 2001; Arc_54, Rolland et al., 2009). The end of the MCA range between 1100 and 1550 AD (Fig. 6b). The majority of the records highlights a transition between warmer and colder periods around the 14[th] century. Two records are characterized by an ending point after the 15[th] century (Arc_49, Linge et al., 2009; Arc_38, Berner et al., 2011). The time coverage of the MCA is about ~200-250 years in most records (Fig. 6c).

The duration and timing of the LIA in the Arctic-subarctic area are more variable from site to site than the MCA, particularly for the starting year (Fig. 7a). The earliest starting point is date around 1200 AD (Esper, 2002; Melvin et al., 2013; Larsen et al., 2011) and the youngest ending point is reported to be as late as 1900 AD (e.g. Gunnarson et al., 2011; Isaksson et al., 2005; Linge et al, 2009, Massa et al., 2012) (Figs. 7a and 7b). The time coverage of the LIA ranges between ~100 years (Kirchhefer, 2001) and ~700 years (Melvin et al., 2013). It does not seems to depend upon the location of the data set in space nor to the

type of archive or proxy (Fig. 7c). The large range of possible timing for the LIA is consistent with the results of previous study in this area (Wanner et al., 2011). It points to difficulty to distinguishing the LIA cooling in subarctic settings. Actually, individual palaeoclimate series from the northern Greenland area did not clearly record the LIA, but a stack of these series highlighted a cold pulse between the 17[th] and 18[th] century (Weissbach et al., 2016). Although the LIA corresponds to negative temperature anomaly, it is difficult to identify the Arctic area solely based on temperature proxies. The evidence of LIA might

also be found in palaeohydrological time series (Nesje and Dahl, 2003). For example, Lamoureux et al. (2001) highlighted the evidence of rainfall increase during the LIA in a varved lake sediment core from the Canadian Arctic. Therefore, it would be relevant to study the LIA from time series sensitive to hydrological variability (Linderholm et al., this issue). This would contribute to a better understanding of secular climate variability in the Arctic area and the role of internal climatic system fluctuations on secular variation during the last millennia.

**4.3. Recent warming and internal climate oscillation**

Studying the climate of the last centuries is a means to examine the important issue of distinguishing the anthropogenic influences from natural variability and the response of ocean/atmosphere coupled system. The last two centuries were characterized in all region by a well-marked warming trend (North Atlantic sector: $\tau$=0.40, p<0.01; Alaska: $\tau$=0.48, p<0.01; Siberia: $\tau$=0.45, p<0.01) (Fig. 8). The temperature increase recorded over the last two centuries is consistent with the increase

of greenhouse gas emissions (Shindell and Faluvegi, 2009). However, the recent warming was not linear as it included different phases of increase highlighted by the 50-years LOESS filtering. This is particularly the case in the subarctic North Atlantic sector, where different periods are distinguished with a pronounced warming transition phase between 1920 and 1930 AD (Fig. 8a). These results suggest the occurrence of multi-decadal variability superimposed on the increasing anthropogenic trend during the last centuries and which can be linked with natural internal climate variability mode,.

In order to determine the origin of the multi-decadal variability in each region, we compared the three regional mean records with two instrumental climate indices: the AMO (Enfield et al., 2001) and the PDO (Mantua et al., 1997), using the wavelet coherence (Figs. 9 and 10, Appendix A). Because one of the main objectives of the paper is to determine the ability of the Arctic 2k database series to mimic the climate variability recorded in the observations data, we did not used the non-

instrumental AMO and PDO records to go father back in time. The analyses were thus performed on the time intervals used to define the AMO and PDO indices, which are 1856-2000 AD and 1900-2000 AD, respectively.

Persistent multi-decadal variability with period of 50-90 years are consistent between the subarctic North Atlantic mean record and the AMO over the last two centuries (1856-2000 AD; Fig. 9c). However, this scale of variability is located in the cone of influence. Comparison of the reconstruction of the 50-90 years oscillation with the original data for each series allowed us to verify if this fluctuation truly characterizes the original signal (Fig. 9a and 9b). It also revealed that fluctuations are in phase and continuous throughout the last two centuries. In the subarctic North Atlantic sector, the 1920-1930 AD transition also coincides with the occurrence of multi-decadal variability with a 20-30 years period similar to the AMO index. Comparison with the instrumental PDO index revealed a 16-30 years oscillation common to the Alaska area and the instrumental index during the 1900-2000 AD interval (Fig. 10b). Wavelet reconstruction of the 16-30 years oscillation for the Alaska palaeoclimate mean record and instrumental PDO index revealed that these scales of fluctuation are in phase. However, while they were continuous throughout the last century for the instrumental index (Fig. 10b), the 16-30 years oscillations only appears after ~1940 AD in the Alaska record (Fig. 10a).

Internal climate fluctuations are also linked with sea-ice cover fluctuations which is an important component of the climate system at high latitude (Miles et al., 2014; Sha et al., 2015, Screen et al., 2016). The relationship between the AMO, the se-ice extent fluctuations and climate variability recorded in the North Atlantic is well-illustrate from 1979 to 2000 (Figure 11). The decline in the sea ice cover was marked by a decrease in the sea ice extend (4% per decade since the end of the 1970's, Cavalieri and Parkinson, 2012), but also ice thickness (50% since 1980 in central Arctic, Kwok and Rothrock, 2009) and the length of the ice season (three-month longer summer ice-free season, Stammerjohn et al., 2012). It was accompanied by heat and moisture transfer to the atmosphere due to the increase of open water surface (Stroeve et al., 2012). This is associated with an increase of surface air temperature, especially in coastal and archipelago areas surrounding the Arctic Ocean (Polyakov et al., 2012). Therefore, while the climate warming in the Arctic accelerates sea ice decline, the sea ice decline simultaneously amplifies and accelerates the recent warming (ice-temperature positive feedback).

Comparison between our three regional mean records and climate index shows the ability of regional proxy-based records to reproduce variability that occurring at multidecadal scales in instrumental data, but also the importance of the role of the internal variability on the climate in the Arctic Area during the last centuries.

## 5. Conclusion

With the publication of the PAGES Arctic 2k database, which contain proxy time series that respond to several quality criteria it was possible to describe the climate in the Arctic-subarctic region over the last 2000 years from low to high frequency variabilities. Long-term tendency, secular variability, but also multidecadal fluctuations with a focus on the last 200 years, were describe using statistical and signal analysis methods.

We presented three new regional mean records for the North Atlantic, the Alaska and the Siberia regions. Due to the uncertainties concerning the relationship between several proxy measurements and instrumental temperatures, climate variability has been studied based on proxy time series rather than temperature reconstructions. A large number of proxy time series in the PAGES Arctic 2k database are from the North Atlantic region, but the Siberia region, and to a lesser extent the Alaska region, are underrepresented. Therefore the global Arctic-subarctic record is probably biased toward the North Atlantic climate variability. Increasing the number of series in the Pacific Arctic, western North America and Siberia would be relevant to gain a better understanding of the global Arctic-subarctic climate variability over the last two millennia.

Despite of the spatial heterogeneity of the database, we found regional long-term tendencies similar to the millennial cooling trend recorded at the global Arctic-subarctic spatial scale, excepted in the Siberia region. Nevertheless, the three regions are characterized by a recent warming starting at the beginning of the 19th century.

Synthesis of the expression of secular fluctuations has shown the spatial and temporal variability of the cold LIA. The definition of the LIA as a major climate event is therefore equivocal, unlike the warm MCA, which seems more evenly represented.

The focus on the last two centuries led to highlight that the recent warming was marked by a global increasing temperature trend linked to the anthropogenic forcing. It was also punctuated by climatic fluctuations related to regional internal climate oscillations that occurring at multidecadal scales, especially the AMO in the North Atlantic region and the PDO in the Alaska region. The identification of this variability in the proxy-based records raise the important issue of the need to better understand regional past climate variability in the Arctic-subarctic area. Comparison between regional proxy-based record and instrumental climate index also lead to propose linkage between paleoclimate series on one side and instrumental data on the other side.

**Acknowledgments**

This is a contribution to the PAGES 2k Network [through the Arctic 2k working group]. Past Global Changes (PAGES) is supported by the US and Swiss National Science Foundations. M.N. was supported by the French Ministry. MD, MN, NM, AD are financed by France Canada research fund and MD, MN and C.C has received funding from the HAMOC project (Grant ANR-13-BS06-0003). We also thanks the FED CNRS 3730: SCALE. We acknowledge all the reviewers for their constructive suggestions and comments.

**Data availability**

The PAGES Arctic 2k database used is this study (v1.1.1) is archived at the National Oceanic and Atmospheric Administration's World Data Center for Paleoclimatology (WDC-Paleo) and available at https://www.ncdc.noaa.gov/paleo/study/16973. The database is also archived on Figureshare and available at https://figshare.com/articles/Arctic_2k_v1_1/1054736/5.

## Appendix A. Wavelet analysis

The Wavelet Transform (WA) is particularly adapted for the study of non-stationary processes, i.e. discontinuities and changes in frequency or magnitude (Torrence and Compo, 1998). Wavelet analysis corresponds to a band-pass filter, which decompose the signal on the base of scaled and translated versions of a reference wave function. Each wavelet has a finite length and is highly localized in time. The reference wavelet $\psi$ comprises two parameters for time-frequency exploration, i.e. scale parameter $a$ and time-localization parameter $b$ so that:

$$\psi_{a,b} = \frac{1}{\sqrt{a}} \, \psi\left(\frac{t-b}{a}\right) \tag{7}$$

The parameter $a$ can be interpreted as a dilation ($a>1$) or contraction ($a<1$) factor of the reference wavelet corresponding to the different scales of observation. The parameter $b$ can be interpreted as a temporal translation or phase shift.

The continuous wavelet transforms of a signal $s(t)$ producing the wavelet spectrum is define as:

$$S_{a,b} = \int_{-\infty}^{+\infty} s(t) \cdot \frac{1}{\sqrt{a}} \cdot \psi\left(\frac{t-b}{a}\right) \cdot dt \tag{8}$$

The so-called local wavelet spectrum allows description and visualization of power distribution (z-axis) according to frequency (y-axis) and time (x-axis).

In this study, the Morlet wavelet was chosen as wavelet reference. Several type of wavelets are available but the Morlet wavelet one offers a good frequency resolution and is most of the time used with a wavenumber of 6 for which wavelet scale and Fourier period are approximately equal.

All series were zero-padded to twice the data length to prevent spectral leakages produced by the finite length of the time series. Zero-padding produces edge effects and the lowest frequencies and near the edges of the series are underestimated. This area is known as the cone of influence. For this reason, fluctuations that occurs in this area have to be interpreted with caution. Detected fluctuations are statistically tested at $\alpha = 0.05$ significance level against an appropriate background spectrum, i.e. a red noise (autoregressive process for AR(1)>0) or a white noise (autoregressive process for AR(1)=0) background (Torrence and Compo, 1998). Autoregressive modelling is used to determine the AR(1) stochastic process for each time series. The detected components can be extracted and reconstructed in the time domain by either inverse Fourier or wavelet transform of selected energy bands in the spectrum.

The cross-wavelet spectrum $W_{XY}(a,T)$ between two signals $x(t)$ and $y(t)$ is calculated according to Eq.(9), where $C_X(a,T)$ and $C^*_Y(a,T)$ are the wavelet coefficient of the signal $x(t)$ and the conjugate of the coefficient of the wavelet of $y(t)$, respectively:

$$W_{XY}(a,T) = C_X(a,T)C^*_Y(a,T) \tag{9}$$

The wavelet coherence is a method that evaluates the correlation between two signals according to the different scales (frequencies) over time. It corresponds to a bivariate extension of wavelet analysis that describes the common variabilities

between two series. The wavelet coherence is analogous to the correlation coefficient between two series in the frequency domain. For two signals $x(t)$ and $y(t)$ se wavelet coherence is calculate as follows:

$$WC(a,T) = \frac{|SW_{XY}(a,T)|}{\sqrt{[|SW_{XX}(a,T).SW_{YY}(a,T)|]}} \tag{10}$$

where $S$ is a smoothing operator.

The wavelet coherence spectrum allows description and visualization of wavelet coherence (z-axis) according to frequency (y-axis) and time (x-axis). Wavelet coherence ranges between 0 and 1, indicating no relationship and a linear relationship between $x(t)$ and $y(t)$, respectively.

Wavelet analysis were performed with the software R (Team, 2008) using the packages biwavelet (Gouhier et al., 2012).

**Appendix B.**

The locally weighted regression (Cleveland and Delvin, 1988; Cleveland and Loader, 1996) was used to investigate systematic features and patterns in the data. It is a method used for smoothing a scatterplot. Contrary to the moving average filtering method, LOESS-filtering allows a well conservation of the analysed signal variance. The polynomial adjustment is locally performed on the whole series of data: a point $x$ is adjusted by the neighbouring points, and weighted by the distance in $x$ of these points. The relative weight of each point depends on its distance of $x$: closer the $x$, the more important is its influence on the shape of the regression, and conversely. For this study, we chose a 50 years windows analysis which allows us to investigate long-term fluctuations but also multi-decadal to centennial variability.

For each individual records a Mann-Kendall test (Mann, 1945 and Kendall, 1975) was used to detect trends in proxy-inferred climate data. It is a non-parametric test commonly employed to detect monotonic trend in climatologic data because it does not require the data to be normally distributed and has low sensitivity to abrupt breaks due to inhomogeneous time series. The null hypothesis H0 is that the data are independent and randomly ordered. The alternative hypothesis H1 is that the data follow a monotonic trend over time. For $n > 10$, the statistic $S$ is approximately normally distributed and positive values of $Z_S$ indicate increasing trends while negative $Z_S$ values show decreasing trends. Testing trends is done at the specific α significance level. When $|Z_S| > |Z_{1-\alpha/2}|$, the null hypothesis is rejected and a significant trend exist in the time series. In this study, significance levels α=0.10, α=0.05 and α = 0.01 were tested. A statistic with which is closely related to $S$ is Kendall's tau. It will take value between -1 and +1. Positive values indicates that the ranks of both variables increase together, so an increasing trend, while a negative correlation indicates a decreasing trend. The closer to +1 or -1 the value of Kendall's tau, the more significant the trend in the time series.

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

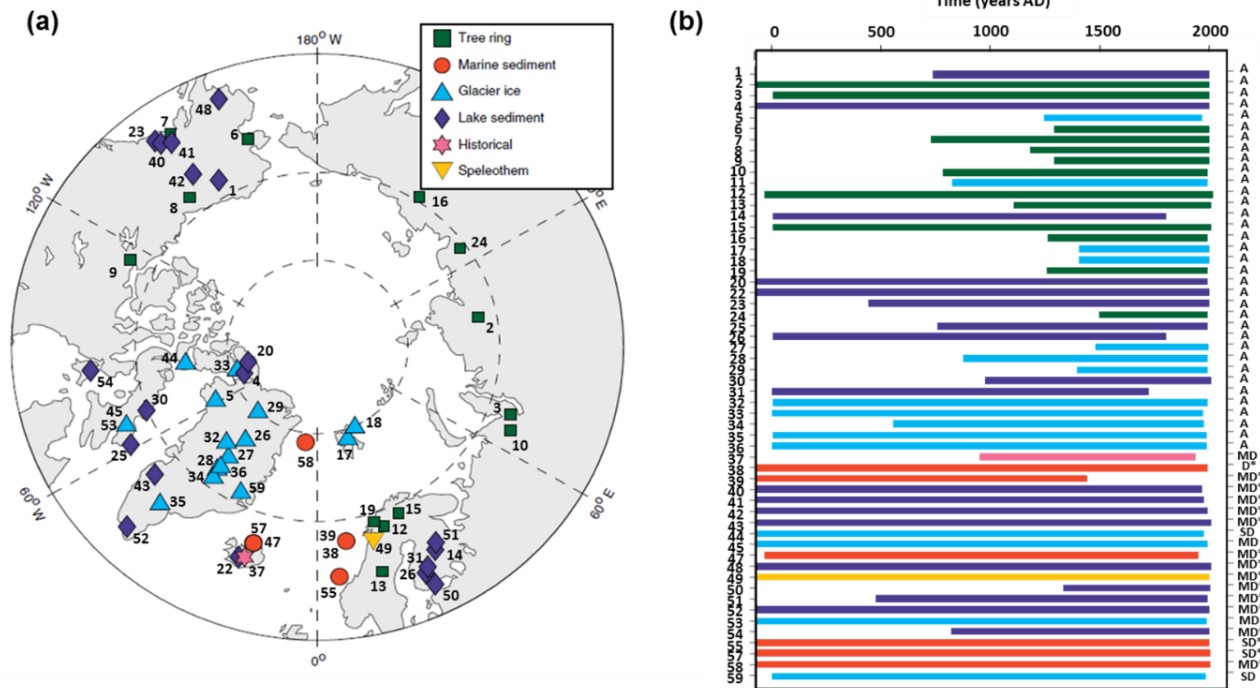

**Figure 1.** Palaeoclimate series used for this study. (a) Polar projection of the proxy records location contained in the PAGES Arctic 2k database (from McKay and Kaufman, 2014). (b) Temporal coverage and resolution (A: annual, SD: Subdecadal, D: Decadal, MD: Multidecadal) of the records from 0 to 2000 AD. Letters with an asterisk indicate a mean temporal resolution. Colours corresponds to archive type and refers to the map legend and numbers in the Arctic 2k database index.

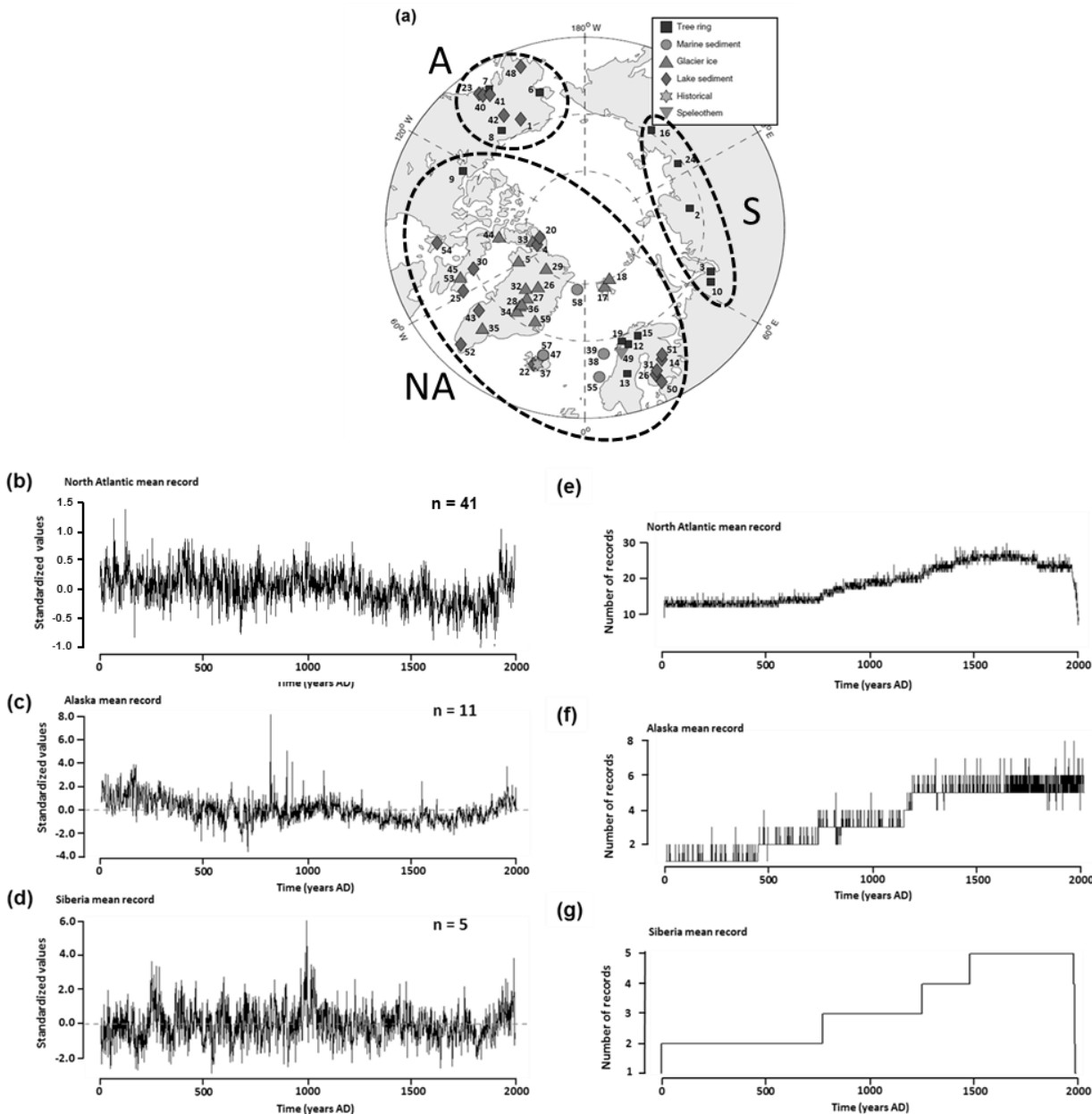

**Figure 2.** (a) Map of record location (modified from McKay and Kaufman, 2014). Dashed lines show selected area used for calculated the three regional mean records (NA: North Atlantic, A: Alaska and S: Siberia) presented at the (b), (c) and (c) curves respectively. n corresponds to the number of records available in each area. Each regional mean record is associated to it corresponding number of records available for each year.

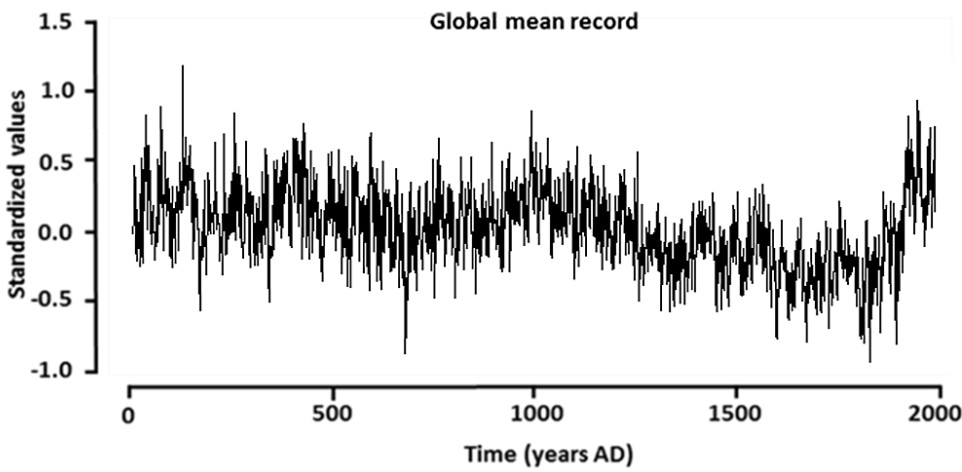

**Figure 3.** Global Arctic mean record obtained from the paleoclimate series contained in the PAGES Arctic 2k database.

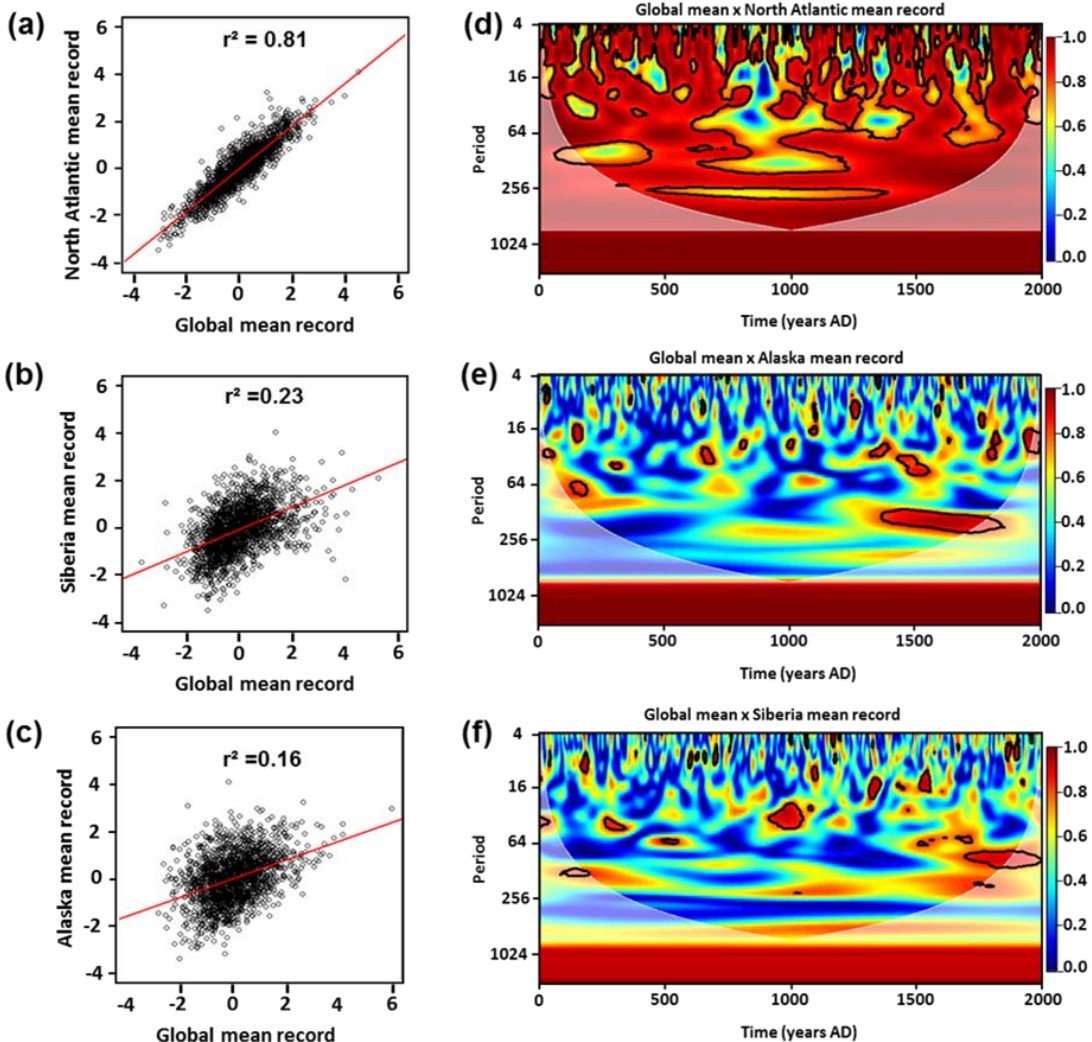

**Figure 4.** *Left.* Correlation between the global mean based on proxy data and the three regional mean records for (a) North Atlantic (b) Alaska and (c) Siberia areas. Correlations is significant at the 95% confidence level. *Right.* (e) Wavelet coherence between global and North Atlantic mean records, (f) global and Alaska mean records, (g) global and Siberia mean records. Colors represent the amplitude of the signal at given time and spectral period (red equals highest power, blue lowest). White line corresponds to cone of influence on wavelet coherence spectrum. Confidence level of 95% ($\alpha$=0.05) is indicated on wavelet spectrum with the black line.

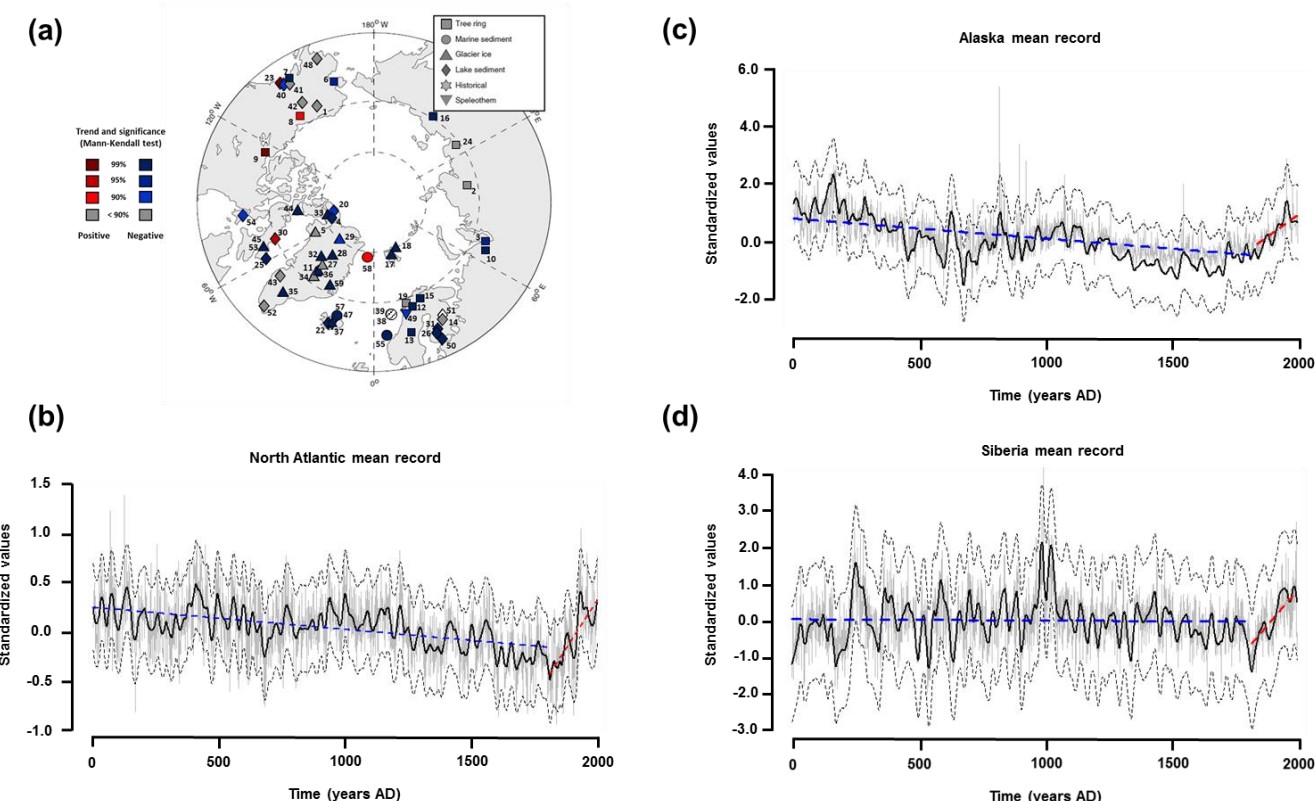

**Figure 5.** (a) Individual trends for each records before recent warming. White dot highlighted inconsistency between two tendencies for the same archive. North Atlantic (b), Alaska (c) and Siberia (d) regional 50-years LOESS. Blue colors indicate decreasing tendency whereas red colors indicate increasing trends. Dashed black lines correspond to the 95% confidence interval.

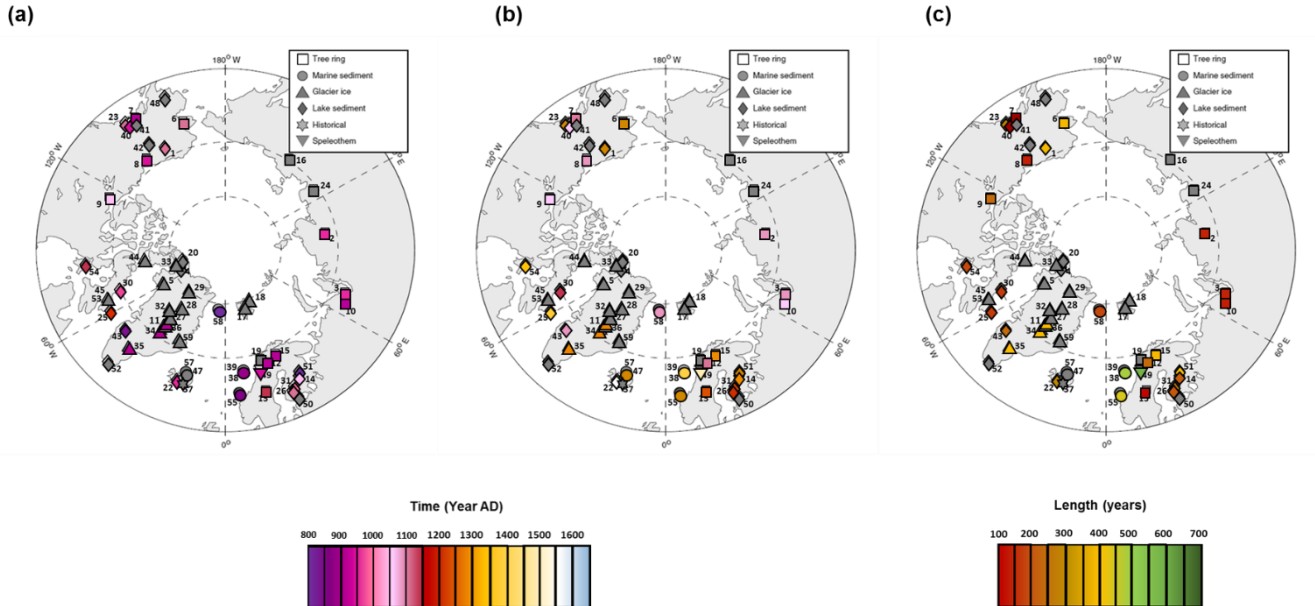

**Figure 6.** Expression of the Medieval Climate Anomaly (MCA) of the Arctic 2k series based on references paper (see McKay and Kaufman, 2014): starting (a), ending (b) and length (c). Symbols in grey correspond to series for which the MCA is not mentioned by the authors. More details concerning the temporal expression of the LIA are available in Table S4 and Figure S2.

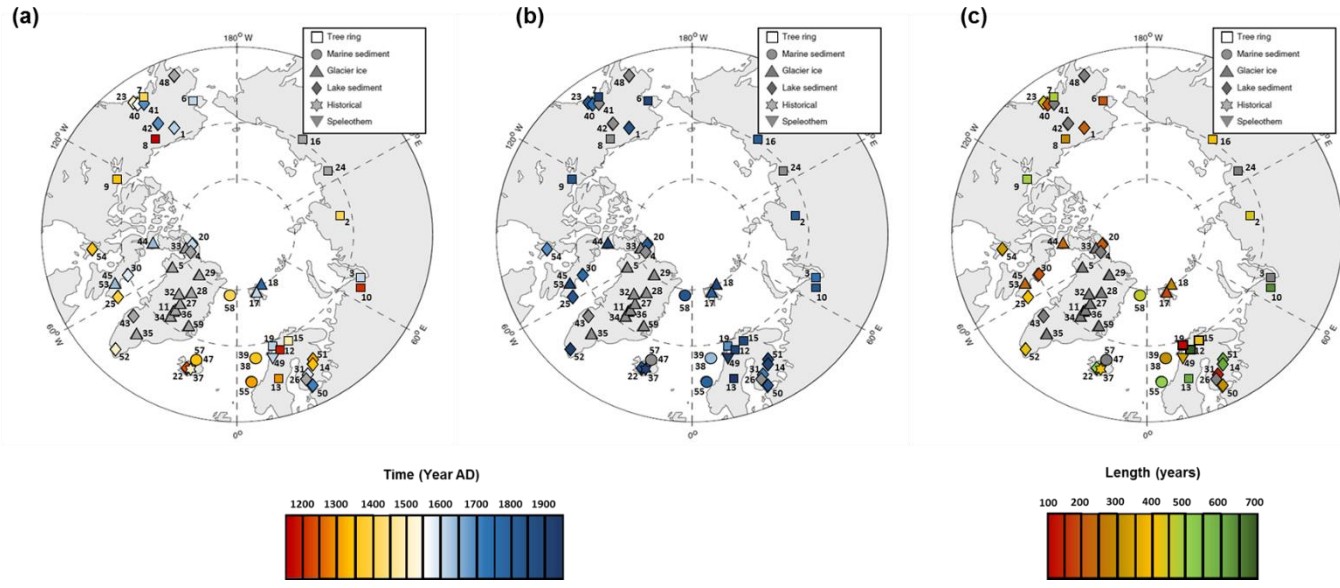

**Figure 7.** Spatial expression of the Little Ice Age (LIA) of the Arctic 2k series based on references paper (see McKay and Kaufman, 2014): starting (a), ending (b) and length (c). Symbols in grey correspond to series for which the LIA is not mentioned by the authors in the original publication. More details concerning the temporal expression of the LIA are available in Table S3 and Figure S1.

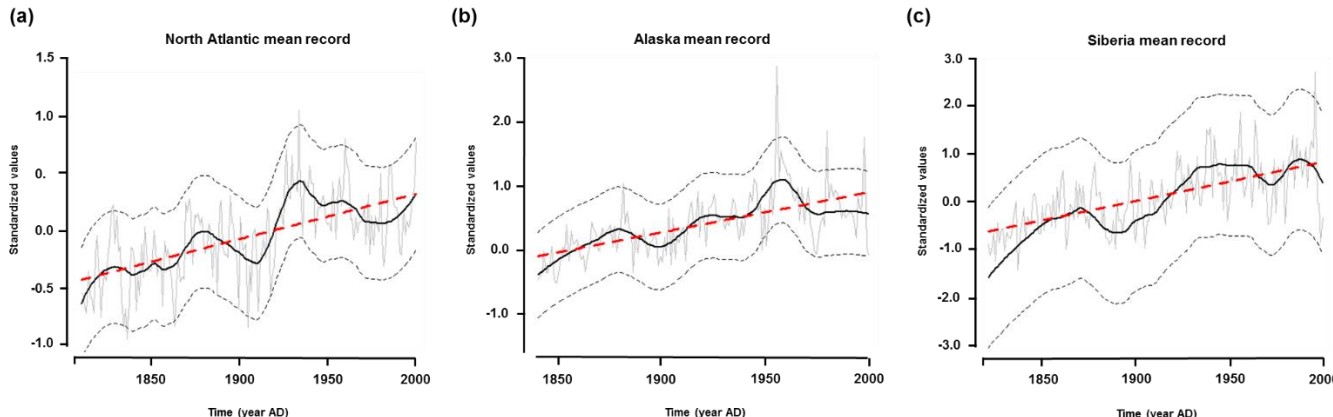

**Figure 8.** Regional mean records of the last two century showing the recent warming period Red dashed lines correspond to linear trend obtained from Mann-Kendall test and black curve to ~50-years loess filtering. Dashed lines correspond to the 95% confidence level interval.

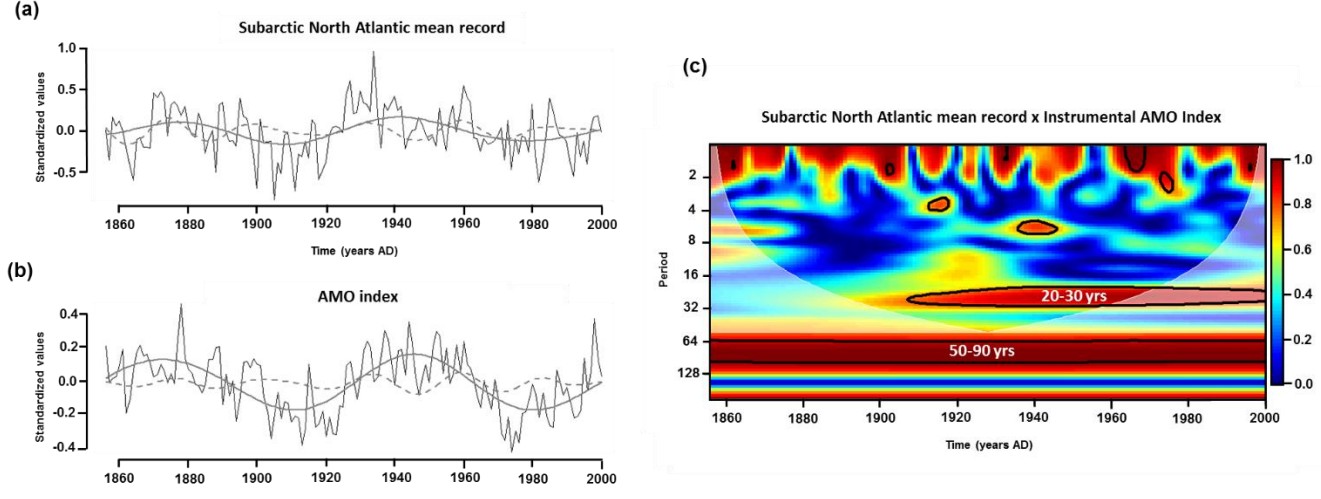

**Figure 9.** Wavelet coherence analysis between subarctic North Atlantic mean record and instrumental AMO index during the 1856-2000 AD period. (a) Subarctic North Atlantic mean record (this study) and (b) instrumental AMO index (Enfield et al., 2001). Grey lines and dashed lines corresponds to filtered and wavelet reconstructed of the ~50-90 years and ~20-30 years periodicities highlighted on wavelet coherence spectrum (c). Colours on coherence wavelet spectrum represent correlation in both time and frequency domain. (Red equals highest correlation and blue lowest). White line corresponds to the cone of influence. Confidence level of 95% (α=0.05) is indicated with the black line.

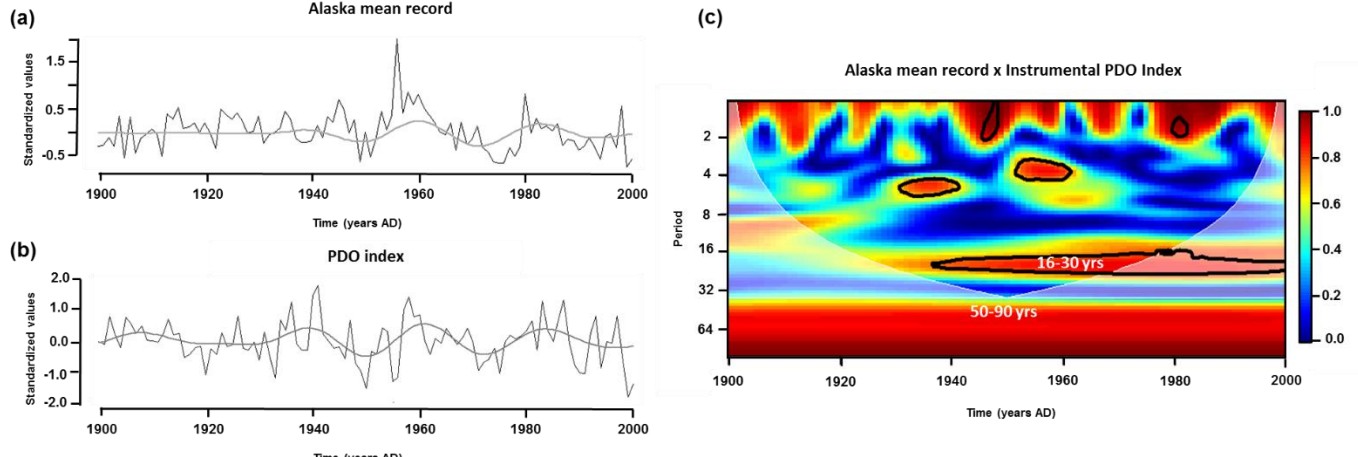

**Figure 10.** Wavelet coherence analysis between Alaska mean record and instrumental PDO index during the 1900-2000 AD period. (a) Alaska mean record (this study) and (b) instrumental PDO index (Mantua et al., 1997). Grey lines corresponds to filtered and wavelet reconstructed of the ~16-30 years periodicity highlighted on wavelet coherence spectrum (c). Colours on coherence wavelet spectrum represent correlation in both time and frequency domain. (Red equals highest correlation and blue lowest). White line corresponds to the cone of influence. Confidence level of 95% ($\alpha=0.05$) is indicated with the black line.

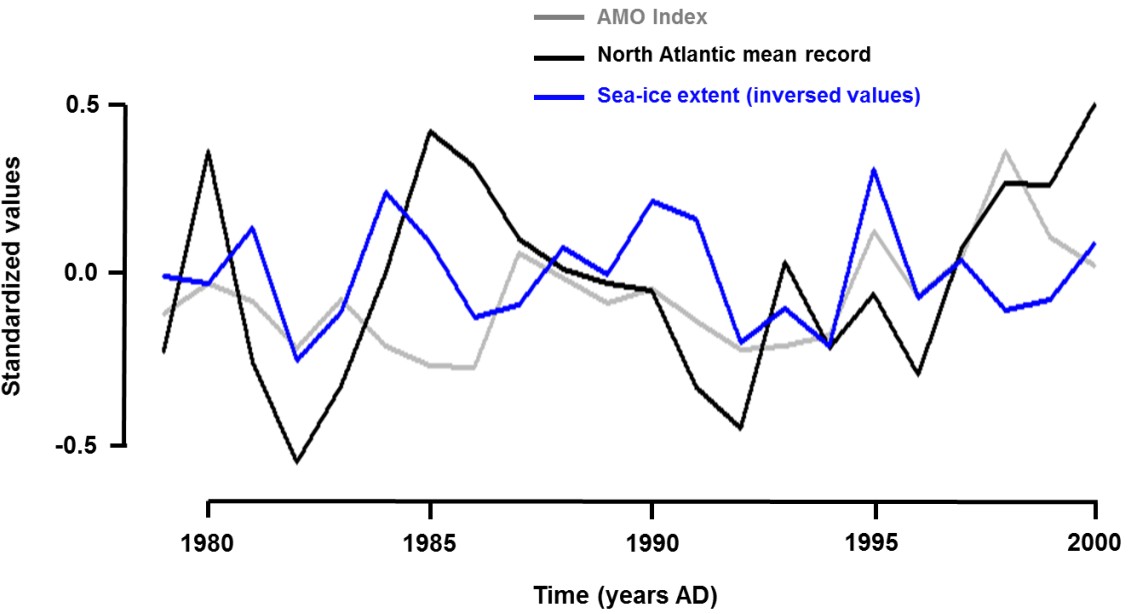

**Figure 11.** Comparison between the North Atlantic mean record based on proxy data, the AMO index and the global Arctic sea-ice extent during their common period (1978-2000).