# Peer review of "Climate variability in subarctic area for the last two millennia"

_Climate of the Past, 2017_

## Referee Comment (RC2) · Anonymous Referee #2 · 29 Apr 2017

Summary

The manuscript by M. Nicolle and others focuses on the climatic covariability in the Arctic, based on the updated Arctic database for proxy records for temperature. The records are examined for temperature trends and the expression of the "Little Ice Age" cold period is presented by referring to the original publications. The records are averaged over three regions, and the resulting composites are analyzed for trends and wavelet coherence with climatic indices. Significant trends, and significant covariabilities are determined. At present, the manuscript requires quite some improvement for language, and, more importantly, a better rigorous treatment of uncertainty to put the trends, and claimed covariabilities, in perspective.

General comments

[Figure]

- The manuscript is generally interesting and relevant.

- There are a lot of spelling problems which, at times, make it hard to understand the meaning of the text.

- There are a lot of literature references throughout the text which, although very interesting, impede the flow. Some of these may be better incorporated in the introduction.

- The manuscript proposes statistical linkages, but does not present them with any form of uncertainty - therefore the claims cannot be validated.

Specific comments

- p1L18: "on many sort" - "on many" or "on multiple proxy types"

- p1L24: "show _a_ relationship"

- p1L25: omit "with"

- p1L31: "temperatures have" or "temperature has".

- p2L5: "lake sediments"

- p2L7: "temperatures"

- p2L16: "by an important spatial and temporal variability expression" - maybe "is expressed spatially and temporally"

- p2L26: "led to"-> "were made to"

- p2L32: "over a large spatial scale"

- p3L2 "based on a regional data set".

- p3L2: "A special attention" - omit A

-p3l8&following - be clear that climatic here means temperature - better replace climatic by temperature where appropriate

- p3l17: assuming that the proxy record climate variability, and that the archiving process does not induce a bias in the multi-annual to centennial frequencies analyzed.

- p4l15- this grouping is, to some extent, arbitrary

-p4l21- regionally

-p4l23- standardizing assumes Gaussianity - is this valid? This may also be problematic when the coverage is not given for the full period

- Do the records that are averaged have, on average, nonzero correlation?

-p5l5 trending tests were done; and what is a "specific" alpha significance level?

-p5l8: with->which

-p6l1 indicate

-p6l6 allows conservation of

-p6l14 decomposes

- Which software packages were used?

- Comment: Figures alongside with the text would be much preferred.

- Fig2: the figure axes labels are on the small side (difficult to read/print)

-p7l21: Consider the possiblity that trends can be due to biases in the proxy e.g. (summer/seasonal bias - Liu et al., PNAS, Rehfeld et al., 2016)

-p8l3 by _a_ OR by decreasing temperatures

-p9l8 remove "date" or say "the earliest start date is ".

-p9l11 it does not appear to

-p9l14 from northern Greenland

-p9l21 I'm assuming you mean "it is difficult to identify it in the Arctic...". But then - how is it defined for other proxies, if not as a cold period - a change? a significant decrease?

-p9l28: issue for

-p9l29 in all regions or in the whole region

-p10l1-5 - Figure 3 – Without uncertianties around these records it is impossible to judge the significance of the trends. Same for Fig. 5

-p9/10: these p-values appear quite low. Are they taking into account autocorrelation and the uncertainty of the composite?

-Fig.6: a,b - uncertainties? What is the assumption on the significance test for the wavelet cross-spectrum? White noise? Red noise?

p10l23-25: Please provide appropriate uncertainty estimates for the wavelet analyses.

p10l25-33 - possibly dedicate a paragraph to sea-ice cover earlier on and show the similarity of the trends.

p11- please have a native speaker correct this manuscript- there are too many mistakes in the final paragraph to correct them in this review.

---

## Referee Comment (RC3) · Anonymous Referee #3 · 3 May 2017

The climate of last 2 k has been very intensively investigated over recent years, and especially since the 2 k datasets were compiled and became available. It is hard to publish paper which would present new, fresh results and/or interpretations. The paper at hand demonstrates this difficulty. It presents the arctic temperature reconstructions based on 56 proxy records, divided into three subregions which are North Atlantic, Alaska, and Siberia. The resulting temperature curves for these subregions are feasible and sound, showing the long-term decline from the beginning of the period until an uptick since the early 19th century. These are the same features as shown in most of the earlier temperature reconstructions for these regions or for larger regions, such as Kaufman et al. (2009) for the whole artic or Hanhijärvi et al. (2013) for the N Atlantic. Of course this is no wonder, as the paper is to great extent based on the same data as the other papers. The only subregion where there is some novelty is Siberia, but

this composite time series is based on only 6 records. I find the new output in these 2000-yr temperature curves minimal.

I also find the chapter "Secular variability" inevitably trivial. First, only the Little Ice Age (LIA) is dealt here, although for example, the Medieval Warm Period or periods between 0 ka to 1 ka could have been included here as well. In the scrutiny, some suggested dates for the beginning and end of LIA are given, and they seem to be compatible with those given in tens of studies about the LIA earlier. Similarly, a set a standard explanations for the cause of LIA are mentioned. So much has been written about the LIA and its causes that one would expect something more substantial than reading that "the LIA has been attributed to the combinations of external climate forcings including solar activity fluctuations and/or volcanic activity". In her book "Little Ice Ages" J.M. Grove (2004) went through possible causes of LIA – 30 pages I notice – discussing solar and volcanic activity and many other possible causes. So one can ask whether the one sentence in the paper at hand adds anything to this issue.

As for the methodological description, it is not necessary to describe the basic mathematics behind the Mann-Kendall test or the wavelet analysis. These are basic techniques in climatology, and if it was necessary to describe the math in this paper, it would be equally necessary in hundreds of other papers where they have been used. The same is true for LOESS, used to provide a smoothers for Fig. 3 – this is a basic smoothing technique, and someone who wants to know more can look at the original paper by Cleveland (1979) – incidentally, this citation is missing from the list of citations in the paper.

To make the paper worth publishing, I suggest that the authors would make a serious effort to find the novel aspects of their dataset and analyses. One way could be to look more at the multidecadal variablity and the covariablity between the temperature trends in the subregions and the atmospheric and ocean oscillations indices, such has been done in chapter "3.3 Recent warming and internal climate oscillations" and in Figs. 6-8. Here the authors compare their temperature records with instrumental AMO

and PDO indices. One possibility would be to use the non-instrumental AMO and PDO records to go farther back in time to see the correlations between multidecadal temperature changes and internal oscillation systems.

To conclude, I would not like to sound discouraging, but it is hard to see enough novelty in the paper to justify publication, unless the authors can substantially expand the new results and aspects of their study.

———————————————————

---

## Short Comment (SC1) · 22 May 2017

The PAGES Data Stewardship Integrative Activity seeks to advance best practices for sharing data generated and assembled as part of all PAGES-related activities. As part of this activity, a team of reviewers has been constituted for the "Climate of the Past 2000 years" Special Issue. The data team is reviewing the data handling within each of the CP-Discussion papers in relation to the CP data policy and current best practices. The team has identified essential and recommended additions for each paper, with the goal of achieving a high and consistent level of data stewardship across the 2k Special Issue. We recognize that an additional effort will likely be required to meet the high level of data stewardship envisaged, and we appreciate the dedication and contribution of the authors. This includes the use of Data Citations (see example in supplement). We ask authors to respond to our comments as part of the regular

open interactive discussion. If you have any questions about PAGES Data Stewardship principles, please contact any of us directly.

Best wishes for the success of your paper,

2k Special Issue Data Review Team (Darrell Kaufman, Nerilie Abram, Belen Martrat, Raphael Neukom, Scott St. George) and ex-officio team members (Marie-France Loutre, Lucien von Gunten)

Essential additions for this paper:

(1) Add a "Data Availability" section with a Data Citation/URL for both the input dataset (PAGES 2k) and one for the primary output from this study. Also, state the source of the code/software used for the wavelet and spectral analyses.

(2) Add Data Citations plus Publication Citations for each of the records used in this study (Table S1). These can be found in version 2 of the global dataset (PAGES 2k Consortium, in press in Scientific Data). In addition, state which of your three regions each of the records is located.

(3) Submit the primary outcome of this study, the composite temperature time series by region (Fig. 2) to a public repository and include the Data Citation/URL in "Data Availability".

(4) Add a statement to explain why the analyses in this study are based on the old version of the PAGES 2k dataset. State how the new and now-discredited data in version 2 of the dataset might influence or limit the conclusions in this study. Alternatively, we highly recommend that the analyses in this study be updated using the current version of the PAGES2k dataset, the same one used by Werner et al. in this special issue

Recommended element:

(5) Archive a table that lists the beginning and ending of the LIA in each record (the data plotted in Fig. 4). These data could also be included in Table S1.

Please also note the supplement to this comment:
http://www.clim-past-discuss.net/cp-2017-33/cp-2017-33-SC1-supplement.pdf
* * *

---

## Author Comment (AC1) · 19 Jun 2017

My co-authors and I would like to thank Dr. D.S. Kaufman for his comments on our submitted manuscript. Here, I would like to response to the essential additions for this paper highlighted by Dr. D.S. Kaufman:

(1) Add a "Data Availability" section: In the updated version of the manuscript, a "Data availability" section is added with the Data URL for the Arctic 2k database v1.1.1 used in this study. As suggested also by the Referee #2, the R software (Team, 2008) used to performed the wavelet analysis and the reference associated to the 'biwavelet' package (Gouhier et al., 2012) used for wavelet analysis will also be added in "2.4. Wavelet Analysis section" of the manuscript.

[Figure]

(2) Add Data Citations and Publication Citations associated to the records: In the updated version of Supplementary Material associated to the manuscript, a new Table is added and replace Table S1. It will contains the description of proxy records in the Arctic 2k database arranged by the three regional regions used in the study, but also reference and it DOI for each record.

(3) Submit the composite temperature time series by region to a public repository: We would like to clarify that we do not produce regional composite temperature time series by region but only regional mean records based on proxy data that has previously been standardized. The regional curves obtained will be published online after the publication of this article.

(4) Why the analyses in this study are based on the old version of the PAGES 2k dataset? In the study, we used the Arctic 2k database v1.1.1 (Mc Kay and Kaufman, 2014) because it was the only version publicly available on the date of our manuscript submission (the February 28th). As you mentioned, a new Arctic 2k database exists but the reference paper associated is not still publicly available (PAGES 2k Consortium, in press; PAGES 2k Consortium, 2017). Without quality criteria of the database it will be difficult to estimate the influence of the use of a new version on the results. Moreover, due to the major difference between the two versions (19 records added and the suppression of 18 records), use the new version does not just correspond to an update of results but means redoing completely the study. This would be very interesting but requires more time and to be the topic of a new paper.

(5) Archive a table that lists the beginning and ending of the LIA: A new Table S3 is added in the Supplementary Material. It will contains the starting and ending dates of the LIA, arranged by the three regional regions used in the study.

References:

Gouhier, T. and Grinsted, A.: biwavelet: Conduct univariate and bivariate wavelet analyses. R package version 0.20.11, Available at: https://cran.r

project.org/web/packages/biwavelet/biwavelet.pdf, 2012.

McKay, N.P. and Kaufman, D.S.: An extended Arctic proxy temperature database for the past 2,000 years, Sci. Data 1:140026, doi: 10.1038/sdata.2014.26, 2014.

PAGES 2k Consortium: A global multiproxy database for temperature reconstructions of the Common Era, Scientific Data, in press.

PAGES 2k Consortium: A global multiproxy database for temperature reconstructions of the Common Era, version 2.0.0, figshare, https://figshare.com/s/d327a0367bb908a4c4f2, 2017.

Team, R. D. C.: R: A language and environment for statistical computing. R Foundation for Statistical Computing, Vienna, Austria. Available at: URL http://www.r-project.org/, 2008.

---

## Author Comment (AC2) · 19 Jun 2017

I would like to thank the Anonymous Reviewer #3 for their comments and suggestions on the submitted manuscript. Here, I would like to respond and provide additional details on the Reviewer #3 comments.

The Anonymous Reviewer #3 note that the description of the Mann-Kendall, LOESS filtering, and wavelet analysis is not necessary for the manuscript. To reduce the "Methods" part, we propose to include it part in several Appendix at the end of the manuscript.

In the new version of the manuscript, the chapter "Secular variability" will be modified. The description of the cold period of the LIA will be developed with the addition of a paragraph on the different ways to characterize the LIA in the Arctic area. The same

synthesis will be made for the warm period of the MCA, with the update of the Fig. 4. The aims of these part are not the definition of forcings that can cause these two major climatic periods and we will focus on the description of the temporal and spatial variability expression of the two periods.

The most important result of our study is the highlighting of variabilities occurring at multidecadal scales record in paleoclimate data and linked to regional internal climate variability observed in instrumental data. So in the update version of the manuscript, the part about the link between climatic oscillation (AMO and PDO) and the proxy data will be developed. Especially, a new figure will be added and presents the similarity between the trends of the AMO index and the sea-ice cover to describe the interaction between internal climate variability, sea-ice cover fluctuations and climate variability recorded in our regional mean records. Because one of the main objectives of the paper is to determine the ability of the Arctic 2k database series to reproduce climate variability recorded in the observations data, we do not use the non-instrumental AMO and PDO records to go farther back in time.

―――――――――――――――――――――

---

## Author Comment (AC3) · 19 Jun 2017

The authors and I would like to thank Reviewer #1 for their comments on our submitted manuscript.

Here, I would like to respond to and provide additional details on the Reviewer comments concerning the grouping into three regions (comments #1, #2 and #3). Since the publication of the Arctic 2k database (McKay and Kaufman, 2014), reconstructions of climate variability obtained from the database was also published at the global Arctic scale or for the Scandinavia region (Linderholm et al., 2015). At this spatial scale, it is difficult to take into account the role of climatic processes on Arctic climate that are known to have regional climatic impacts (e.g. climatic oscillations as AMO or PDO).

We choose a regional approach to refine the comprehension of climate variability in the Arctic area and the three group was determined based on the regional impact of climatic oscillation found in the literature. We agree that using an EOF analysis will be a good way to find the major patterns existing into the database. However, one of the main objectives of the paper is to determine the ability of the Arctic 2k database series to reproduce climate variability recorded in the observations data, especially during their common period. So that's why we choose to determine our regions based on the regional effects of internal atmosphere/ocean oscillations on climate that are currently recorded in instrumental data and not based on variability recorded in the palaeoclimate series. Regional effect of internal atmosphere/ocean oscillations on climate are described in the manuscript (p4. L2-13).

Reviewer #1 also suggests developing the part concerning the internal climate variability (comment #4). In fact, one of the most important results of our study is the highlighting of variabilities occurring at multidecadal scales record in paleoclimate data and linked to regional internal climate variability observed in instrumental data. We agree that the use of the recent observations allows us to determine the pattern of influence of PDO and AMO on climate in our study area but also including the role of the sea-ice cover variability. Using recent observations also allow us the compare it with our three regional mean records in order to determine the ability of them to reproduce the climate variability observed.

References

Linderholm H.W., Björklund J., Seftigen K., Gunnarson B.E. and Fuentes M.: Fennoscandia revisited: a spatially improved tree ring reconstruction of summer temperatures for the last 900 years, Climate Dynamics, 45(3), 933-947, doi:10.1007/s00382-014-2328-9 2015.

McKay, N.P. and Kaufman, D.S.: An extended Arctic proxy temperature database for the past 2,000 years, Sci. Data 1:140026, doi: 10.1038/sdata.2014.26, 2014.

---

## Author Comment (AC4) · 19 Jun 2017

My co-authors and I would like to thank the Anonymous Reviewer #2 for their comments and suggestions on our submitted manuscript. Here, I would like to provide additional details on the Reviewer #2 comments concerning a better rigorous treatment of uncertainty to put the trends and claimed covariabilities between our regional mean records and the instrumental climate index. For a better understanding, language comments are dissociated for the rest of the specific comments.

Specific comments:

- One of the comments of the Reviewer #2 is to be clear when climatic means temperatures". In the update version of the manuscript, "Climatic" was modified where it's

appropriate.

- The Reviewer #2 reports an arbitrary regional grouping of the series. The spatial density of the data set was the first argument to group the series into three regions, but the regional impact of climatic oscillation observed in instrumental data and found the literature allows to justify that this grouping has a currently climatic reality. In fact, one of the main objectives of the paper is to determine the ability of the Arctic 2k database series to reproduce the regional internal variability recorded in the observations data. Maybe this objective is not clearly noticed in the introduction of the paper and we have to insist on that.

- p4L23: Standardization does not change the underlying distribution. It only changes the units. It is used here to compare the proxy series with different units. All the series were individually standardized before calculate each regional mean records.

- The alpha ($\alpha$) significance level is the probability of rejecting the null hypothesis - data are independent and randomly ordered – and a significant trend exists. The term 'specific' is not really appropriate and it will be removed in the updated version of the manuscript.

- Concerning the software packages used, we propose to add the following sentence into the 2.4 Wavelet analysis section is add in the update version of the manuscript: 'Wavelet analysis were performed with the software R (Team, 2008) using the packages biwavelet (Gouhier et al., 2012).'

- The size of the axes labels on Fig. 2 is increase for a better readability.

- The Reviewer #2 suggest being more specific to explain the difficulty in identifying the Little Ice Age in the Arctic. A paragraph describing the different expressions of the LIA in Arctic will be added to the updated version of the manuscript. A new paragraph will also be added and the same synthesis will be made for the warm period of the MCA, with the update of the Fig. 4.

- It is noticeable that without uncertainties around regional mean records presented in Fig. 3 and Fig. 5, it is impossible to judge the significance of the trends. In the updated version of the manuscript Fig. 3 and Fig. 5, but also Fig. 6, will be modified and the standard deviation curves for the three regional mean records will be added. The addition of the number of records used for each regional mean records and the standard deviation is sufficient to evaluate the uncertainty around regional mean records and so the significance of the trends.

- p9 and 10: Indeed, a p-value of 0.05 is commonly used for a statistical test. If trend detected in the regional mean records are significant at 99% confidence level (p-value < 0.01), it is also true for a 95% confidence level (p-value < 0.05). Given that the 95% confidence level is used for the wavelet analysis, we will change this value to homogenize the new version of the manuscript. Autocorrelation and partial autocorrelation was calculated for the three regional mean records (see Fig.1 below). Results show that Mann-Kendall trend detected are not linked to autocorrelation in the regional mean records generated.

- Concerning the uncertainty estimation for the wavelet analysis, it is included in the statistical test associated. For all local wavelet spectra, the statistical significance of peaks is assessed using Monte Carlo simulation against an appropriate background noise. Autoregressive modeling is used to determine the AR(1) stochastic process for each time series. AR(1) background noise could be either a red noise (AR(1)>0) or a white noise (AR(1)=0). Each AR(1) is calculated before performed wavelet analysis to determine the background noise used. Concerning the cross-wavelet spectrum, detected fluctuations are statistically tested at $\alpha = 0.05$ significance level against a red noise background. All these precision will be added in the 2.4. Wavelet Analysis part of the manuscript.

- In the update version of the manuscript, the part about the link between climatic oscillation (AMO and PDO) and the proxy data will be developed, including the role of sea-ice cover on climate. Especially, a new figure will be added and presents the

similarity between the trends of the AMO index and the sea-ice cover.

Language comments:

All the language mistakes listed by the Anonymous Reviewer #2 were taken into account. Specifically:

- p1L18: replaced 'on many sorts of proxy data' with 'on multiple proxy type records'

- p1L24: replaced 'show relationship' with 'show a relationship'

- p1L31: replaced 'temperature have' with 'temperature has'

- p2L5: added 's' to 'lake sediments'

- p2L7: added 's' to 'temperatures'

- p2L16: replaced the sentence 'The LIA is however characterized by an important spatial and temporal variability expression, particularly visible at more regional scale (e.g. Pages 2k Consortium, 2013).' with 'The LIA is known to have an important spatial and temporal variability, particularly at regional scale (e.g. Pages 2k Consortium, 2013).'

- p2L26: replaced 'led to' with 'were made to'

- p2L32: added 'a' to 'over a large spatial scale'

- p3L2: replaced 'data sets' with 'data set'

- p3L2: removed 'a' to 'A special attention'

- p4L21: replaced 'regional' with 'regionally'

- p5L5: removed 's' to 'trends'

- p5L8: replaced 'with' with 'which'

- p6L1: removed 's' to 'indicates'

- p6L6: replaced 'well-conservation' with 'conservation'

- p6L14: added 's' to 'decomposes'

- p8L3: replaced 'pronounced decreasing trend of temperatures' with 'decreasing temperatures'

- p9L8: removed 'date'

- p9L11: replaced 'seems' with 'appear'

- p9L14: 'removed 'the'

- p9L28: replaced 'to' with 'for'

- p9L29: added 's' to 'regions'

- The final paragraph will be entirely corrected.
* * *
**Series Atlantic_mean_record**

**Series Alaska_mean_record**

**Series Siberia_mean_record**

**Fig. 1.** Partial autocorrelation calculated for the three regional mean records

---

## Author Response (AR1)

The authors and I would like to thank Reviewer #1 for their comments on our submitted manuscript.

Here, I would like to respond to and provide additional details on the Reviewer comments concerning the grouping into three regions (**comments #1, #2 and #3**).
Since the publication of the Arctic 2k database (McKay and Kaufman, 2014), reconstructions of climate variability obtained from the database were published either at the global arctic scale either for a specific region (Scandinavia) (Linderholm *et al.*, 2015). This both spatial scale do not allowed to take into account the role of climatic processes on arctic climate that are well known to have regional climatic impacts nowadays (*e.g.* climatic oscillations as AMO or PDO). We choose a regional approach to refine the comprehension of climate variability in the Arctic area and the three group were determined based on the regional impact of climatic oscillation found in the literature.
We agree that using an EOF analysis would be a good way to find the major patterns existing into the database. However, one of the main objectives of the paper is to determine the ability of the Arctic 2k database series to reproduce climate variability recorded in the observations data, especially during their common period. So that's why we choose to determine our regions based on the regional effect of internal atmosphere/ocean oscillations on climate that are currently recorded in instrumental data and not based on variability recorded in the palaeoclimate series. Regional effect of internal atmosphere/ocean oscillations on climate are describe in the manuscript (p4. L2-13).

Reviewer #1 also suggest to develop the part concerning the internal climate variability (**comment #4**). In fact, one of the most important result of our study is to highlight of the variabilities occurring at multidecadal scales record in paleoclimate data and linked to regional internal climate variability observed in instrumental data. We agree that the use of the recent observations allows us to determine the pattern of influence of PDO and AMO on climate in our study area but also including the role of the sea-ice cover variability. Using recent observations also allow us to compare it with our three regional mean records in order to determine the ability of them to reproduce the climate variability observed.

**References**

Linderholm H.W., Björklund J., Seftigen K., Gunnarson B.E. and Fuentes M.: Fennoscandia revisited: a spatially improved tree-ring reconstruction of summer temperatures for the last 900 years, Climate Dynamics , 45(3), 933-947, doi:10.1007/s00382-014-2328-9 2015.

McKay, N.P. and Kaufman, D.S.: An extended Arctic proxy temperature database for the past 2,000 years, Sci. Data 1:140026, doi: 10.1038/sdata.2014.26, 2014.

My co-authors and I would like to thank the anonymous Reviewer #2 for their comments and suggestions on our submitted manuscript. Here, I would like to provide addition details on the Reviewer #2 comments concerning a better rigorous treatment of uncertainty to put the trends and claimed covariabilities between our regional mean records and the instrumental climate index. For a better understanding, language comments are dissociated for the rest of the specific comments.

**Specific comments:**

- One of the comments of the Reviewer #2 is to be clear when **climatic means temperatures**". In the update version of the manuscript "Climatic" was modified where it's appropriate.

- The Reviewer #2 reports an arbitrary **regional grouping of the series**. The spatial density of the data set was the first argument to group the series into three regions, but the regional impact of climatic oscillation observed in instrumental data and found the literature allows to justify that this grouping has a currently climatic reality. In fact, one of the main objectives of the paper is to determine the ability of the Arctic 2k database series to reproduce the regional internal variability recorded in the observations data. May be this objective is not clearly notice in the introduction of the paper and we have to insist on that.

- p4L23**: Standardization** does not change the underlying distribution. It only changes the units. It is used here to compare the proxy series with different units. All the series were individually standardized before calculate each regional mean records.

- The **alpha (α) significance level** is the probability of rejecting the null hypothesis - data are independent and randomly ordered – and a significant trend exist. The term 'specific' is not really appropriate and it will be removed in the updated version of the manuscript.

- Concerning the **software packages used**, we propose to add the following sentence into the 2.4 Wavelet analysis section is add in the update version of the manuscript: 'Wavelet

analysis were performed with the software R (Team, 2008) using the packages biwavelet (Gouhier et al., 2012).'

- The size of the **axes labels on Fig. 2** are increase for a better readability.

- The Reviewer #2 suggest to **be more specific to explain the difficulty to identify the Little Ice Age** in the Arctic. A paragraph describing the different expressions of the LIA in Arctic will be added to the updated version of the manuscript. A new paragraph will also be added and the same synthesis will be made for the warm period of the MCA, with the update of the Fig. 4.

- It is notice that without uncertainties around regional mean records presented in Fig. 3 and Fig. 5, it is impossible to judge the **significance of the trends**. In the updated version of the manuscript Fig. 3 and Fig. 5, but also Fig. 6, will be modified and the standard deviation curves for the three regional mean records will be added. The addition of the number of records used for each regional mean records and the standard deviation is sufficient to evaluate the uncertainty around regional mean records and so the significance of the trends.

- p9 and 10: Indeed, **p-value** of 0.05 is commonly used for statistical test. If trend detected in the regional mean records are significant at 99% confidence level (p-value $< 0.01$), it is also true for a 95% confidence level (p-value $< 0.05$). Given that the 95% confidence level is used for the wavelet analysis, we will change this value to homogenize the new version of the manuscript. Autocorrelation and partial autocorrelation was calculated for the three regional mean records (see figures below). Results show that Mann-Kendall trend detected are not linked to autocorrelation in the regional mean records generated.

[Figure]

[Figure]

[Figure]

- Concerning the **uncertainty estimation for the wavelet analysis**, it is includes in the statistical test associated. For all local wavelet spectra, the statistical significance of peaks is assessed using Monte Carlo simulation against an appropriate background noise. Autoregressive modelling is used to determine the AR(1) stochastic process for each time series. AR(1) background noise could be either a red noise (AR(1)>0) or a white noise (AR(1)=0). Each AR(1) is calculated before performed wavelet analysis to

determine the background noir used. Concerning the cross-wavelet spectrum, detected fluctuations are statistically tested at α = 0.05 significance level against a red noise background. All these precision will be added in the 2.4. Wavelet Analysis part of the manuscript.

- In the update version of the manuscript the part about the link between climatic oscillation (AMO and PDO) and the proxy data will be developed, including the **role of sea-ice cover** on climate. Especially, a new figure will be added and presents the similarity between the trends of the AMO index and the sea-ice cover.

**Language comments:**

All the language mistakes listed by the anonymous Reviewer #2 were taken into account. Specifically:

- p1L18: replaced 'on many sort of proxy data' with 'on multiple proxy type records'
- p1L24: replaced 'show relationship' with 'show a relationship'
- p1L31: replaced 'temperature have' with 'temperature has'
- p2L5: added 's' to 'lake sediments'
- p2L7: added 's' to 'temperatures'
- p2L16: replaced the sentence 'The LIA is however characterized by an important spatial and temporal variability expression, particularly visible at more regional scale (e.g. Pages 2k Consortium, 2013).' with 'The LIA is known to have an important spatial and temporal variability, particularly at regional scale (e.g. Pages 2k Consortium, 2013).'
- p2L26: replaced 'led to' with 'were made to'
- p2L32: added 'a' to 'over a large spatial scale'
- p3L2: replaced 'data sets' with 'data set'
- p3L2: removed 'a' to 'A special attention'
- p4L21: replaced 'regional' with 'regionally'
- p5L5: removed 's' to 'trends'
- p5L8: replaced 'with' with 'which'
- p6L1: removed 's' to 'indicates'
- p6L6: replaced 'well-conservation' with 'conservation'
- p6L14: added 's' to 'decomposes'

- p8L3: replaced 'pronounced decreasing trend of temperatures' with 'decreasing temperatures'
- p9L8: removed 'date'
- p9L11: replaced 'seems' with 'appear'
- p9L14: 'removed 'the'
- p9L28: replaced 'to' with 'for'
- p9L29: added 's' to 'regions'
- The final paragraph was entirely corrected.

I would like to thank the anonymous Reviewer #3 for their comments and suggestions on the submitted manuscript. Here, I would like to response and provide additional details on the Reviewer #3 comments.

The anonymous Reviewer #3 note that the description of the Mann-Kendall, LOESS filtering and wavelet analysis is not necessary in the manuscript. To reduce the "**Methods" part**, we propose to include it part in several Appendix at the end of the manuscript.

In the new version of the manuscript, the **chapter "Secular variability"** will be modified. The description of the cold period of the LIA will be develop with the addition of a paragraph on the different way of characterizing the LIA in the Arctic area. The same synthesis will be made for the warm period of the MCA, with the update of the Fig. 4. The aims of these part is not the definition of forcings that can cause these two major climatic period and we will focus on the description of the temporal and spatial variability expression of the two periods.

The most important result of our study is the highlighting of variabilities occurring at multidecadal scales record in paleoclimate data and linked to regional internal climate variability observed in instrumental data. So in the update version of the manuscript the part about the link between climatic oscillation (AMO and PDO) and the proxy data will be developed. Especially, a new figure will be added and presents the similarity between the trends of the AMO index and the sea-ice cover to describe the interaction between internal climate variability, sea-ice cover fluctuations and climate variability record in our regional mean records. Because one of the main objectives of the paper is to determine the ability of the Arctic 2k database series to reproduce climate variability recorded in the observations data, we do not used the non-instrumental AMO and PDO records to go father back in time.

My co-authors and I would like to thank Dr. D.S. Kaufman for his comments on our submitted manuscript. Here, I would like to response to the essential additions for this paper highlighted by Dr. D.S. Kaufman:

**(1) Add a "Data Availability" section** :

In the updated version of the manuscript, a "Data availability" section is added with the Data URL for the Arctic 2k database v1.1.1 used in this study.

As suggested also by the Referee #2, the R software (Team, 2008) used to performed the wavelet analysis and the reference associated to the 'biwavelet' package (Gouhier et al., 2012) used for wavelet analysis will also be added in "2.4. Wavelet Analysis section" of the manuscript.

**(2) Add Data Citations and Publication Citations associated to the records** :

In the updated version of Supplementary Material associated to the manuscript, a new Table is added and replace Table S1. It will contains the description of proxy records in the Arctic 2k database arranged by the three regional regions used in the study, but also reference and it DOI for each record.

**(3) Submit the composite temperature time series by region to a public repository** :

We would like to clarify that we do not produce regional composite temperature time series by region but only regional mean records based on proxy data that has previously been standardized. The regional curves obtained will be published online after the publication of this article.

**(4) Why the analyses in this study are based on the old version of the PAGES 2k dataset?**

In the study, we used the Arctic 2k database v1.1.1 (Mc Kay and Kaufman, 2014) because it was the only version publicly available on the date of our manuscript submission (the February 28th). As you mentioned, a new Arctic 2k database exists but the reference paper associated is not still publicly available (PAGES 2k Consortium, *in press*; PAGES 2k Consortium, 2017). Without quality criteria of the database it will be difficult to estimate the influence of the use of a new version on the results. Moreover, due to the major difference between the two versions (19 records added and the suppression of 18 records), use the new version does not just correspond to an update of results but means redoing completely the study. This would be very interesting but requires more time and to be the topic of a new paper.

**(5) Archive a table that lists the beginning and ending of the LIA :**

A new Table S3 is added in the Supplementary Material. It will contains the starting and ending dates of the LIA, arranged by the three regional regions used in the study.

List of relevant changes made in the manuscript:

- **The introduction of the manuscript was modified** in order to better explain the objective of the paper: (1) study of the regional climate variability in the Arctic-subarctic area, (2) determine the ability of the proxy records to reproduce instrumental climate variability and (3) role of the internal climate variability during the last two centuries. **The conclusion was also rewrite**.

- To **reduce the** "**Methods" part**, the description of the Mann-Kendall, LOESS filtering and wavelet analysis was include in **several Appendix** at the end of the manuscript.

**- A new part presenting comparison between a global Arctic mean record and the three regional curves was added and developed** to add precision concerning the grouping into three groups. The choice of regional approach is now explain by the spatial distribution of the serie and the groupping is justified by actual regional climatic influence.

- **Confidence interval were add to judge the significance of the LOESS-filtering and the trend in Figures 5 and 8.**

- **The "Secular variability part" was developed** with a synthesis of the expression of the MCA in the Arctic-subarctic region and **a new figure was added** (Fig. 6). **Two supplementary Tables and Figures was also added** (beginning and ending of the LIA and the MCA)

- **A new figure was added and presents the similarity between the trends of the AMO index and the sea-ice cover** to describe the interaction between internal climate variability, sea-ice cover fluctuations and climate variability record in our regional mean records.

[revised manuscript text omitted]

**Commenté [NA1]:** Modified in order to better explain the objective of the paper: (1) study of the regional climate variability in the Arctic-subarctic area, (2) determine the ability of the proxy records to reproduce instrumental climate variability and (3) role of the internal climate variability during the last two centuries – Referee #1, Referee #2 and Referee #3

To put the present Arctic warming in perspective against the natural climate variability, the instrumental time series are not sufficient. It is thus necessary to extend the climate record back in time with proxy data measured in palaeoclimate archives such as tree rings, ice cores, lake sediment etc.... to help distinguishing the anthropogenic influences from natural forcings (e.g. solar activity, volcanism) and the internal response of the ocean/atmosphere coupled system. Continental multi-proxy reconstructions reveal declining temperat...

**Commenté [NA2]:** The analyses are based on the v1.1.1 PAG...

**Commenté [NA3]:** Table S1 was replace by a new table S1

[revised manuscript text omitted]

Commenté [NA5]: A new table S3 was added in the supplementary material and contains the starting and ending dates arranged by three regions. – Comment (5) – D.s Kaufman

MCA started at the end of the 12th century (Arc_25, Moore et al., 2001; Arc_54, Rolland et al., 2009). The end of the MCA range between 1100 and 1550 AD (Fig. 6b). The majority of the records highlights a transition between warmer and colder periods around the 14th century. Two records are characterized by an ending point after the 15th century (Arc_49, Linge et al., 2009; Arc_38, Berner et al., 2011). The time coverage of the MCA is about ~200-250 years in most records (Fig. 6c).

5 The duration and timing of the LIA in the Arctic-subarctic area are more variable from site to site than the MCA, particularly for the starting year (Fig. 7a). The earliest starting point is date around 1200 AD (Esper, 2002; Melvin et al., 2013; Larsen et al., 2011) and the youngest ending point is reported to be as late as 1900 AD (e.g. Gunnarson et al., 2011; Isaksson et al., 2005; Linge et al, 2009, Massa et al., 2012) (Figs. 7a and 7b). The time coverage of the LIA ranges between ~100 years (Kirchhefer, 2001) and ~700 years (Melvin et al., 2013). It does not seems to depend upon the location of the data set in space nor to the
10 type of archive or proxy (Fig. 7c). The large range of possible timing for the LIA is consistent with the results of previous study in this area (Wanner et al., 2011). It points to difficulty to distinguishing the LIA cooling in subarctic settings. Actually, individual palaeoclimate series from the northern Greenland area did not clearly record the LIA, but a stack of these series highlighted a cold pulse between the 17th and 18th century (Weissbach et al., 2016). Although the LIA corresponds to negative temperature anomaly, it is difficult to identify the Arctic area solely based on temperature proxies. The evidence of LIA might
15 also be found in palaeohydrological time series (Nesje and Dahl, 2003). For example, Lamoureux et al. (2001) highlighted the evidence of rainfall increase during the LIA in a varved lake sediment core from the Canadian Arctic. Therefore, it would be relevant to study the LIA from time series sensitive to hydrological variability (Linderholm et al., this issue). This would contribute to a better understanding of secular climate variability in the Arctic area and the role of internal climatic system fluctuations on secular variation during the last millennia.

20 **4.3. Recent warming and internal climate oscillation**

Studying the climate of the last centuries is a means to examine the important issue of distinguishing the anthropogenic influences from natural variability and the response of ocean/atmosphere coupled system. The last two centuries were characterized in all region by a well-marked warming trend (North Atlantic sector: $\tau$=0.40, p<0.01; Alaska: $\tau$=0.48, p<0.01; Siberia: $\tau$=0.45, p<0.01) (Fig. 8). The temperature increase recorded over the last two centuries is consistent with the increase
25 of greenhouse gas emissions (Shindell and Faluvegi, 2009). However, the recent warming was not linear as it included different phases of increase highlighted by the 50-years LOESS filtering. This is particularly the case in the subarctic North Atlantic sector, where different periods are distinguished with a pronounced warming transition phase between 1920 and 1930 AD (Fig. 8a). These results suggest the occurrence of multi-decadal variability superimposed on the increasing anthropogenic trend during the last centuries and which can be linked with natural internal climate variability mode.
30 In order to determine the origin of the multi-decadal variability in each region, we compared the three regional mean records with two instrumental climate indices: the AMO (Enfield et al., 2001) and the PDO (Mantua et al., 1997), using the wavelet coherence (Figs. 9 and 10, Appendix A). Because one of the main objectives of the paper is to determine the ability of the Arctic 2k database series to mimic the climate variability recorded in the observations data, we did not used the non-

Commenté [NA6]: Added to also describe the expression of the warm MCA in the Arctic area – Referee #3

[revised manuscript text omitted]

Helama, S., Fauria, M. M., Mielikainen, K., Timonen, M. and Eronen, M.: Sub-Milankovitch solar forcing of past climates: Mid and late Holocene perspectives, Geol. Soc. Am. Bull, 122, 1981–1988, 2010. ¶

[revised manuscript text omitted]

Commenté [NA11]: Added to precise concerning the way of grouping the data into three groups. The regional approach is now explain by the spatial distribution of the serie and the groupping is confirmed by actual regional climatic influence - Comments (1), (2) and (3) - Referee#1

[Figure]

**Figure 4.** *Left.* Correlation between the global mean based on proxy data and the three regional mean records for (a) North Atlantic (b) Alaska and (c) Siberia areas. Correlations is significant at the 95% confidence level. *Right.* (e) Wavelet coherence between global and North Atlantic mean records, (f) global and Alaska mean records, (g) global and Siberia mean records. Colors represent the amplitude of the signal at given time and spectral period (red equals highest power, blue lowest). White line corresponds to cone of influence on wavelet coherence spectrum. Confidence level of 95% (α=0.05) is indicated on wavelet spectrum with the black line.

**Commenté [NA12]:** Developed to precise concerning the way of grouping the data into three groups. The regional approach is now explain by the spatial distribution of the serie and the groupping is confirmed by actual regional climatic influence - Comments (1), (2) and (3) - Referee#1

[Figure]

**Figure 5.** (a) Individual trends for each records before recent warming. White dot highlighted inconsistency between two tendencies for the same archive. North Atlantic (b), Alaska (c) and Siberia (d) regional 50-years LOESS. Blue colors indicate decreasing tendency whereas red colors indicate increasing trends. Dashed black lines correspond to the 95% confidence interval.

**Commenté [NA13]:** Confidence interval were add to judge the significance of the LOESS-filtering and the trend – Referee #2

[Figure]

**Figure 6.** Expression of the Medieval Climate Anomaly (MCA) of the Arctic 2k series based on references paper (see McKay and Kaufman, 2014): starting (a), ending (b) and length (c). Symbols in grey correspond to series for which the MCA is not mentioned by the authors. More details concerning the temporal expression of the LIA are available in Table S4 and Figure S2.

Commenté [NA14]: Added to also describe the expression of the warm MCA in the Arctic area – Referee #3

[Figure]

**Figure 7.** Spatial expression of the Little Ice Age (LIA) of the Arctic 2k series based on references paper (see McKay and Kaufman, 2014): starting (a), ending (b) and length (c). Symbols in grey correspond to series for which the LIA is not mentioned by the authors in the original publication. More details concerning the temporal expression of the LIA are available in Table S3 and Figure S1.

[Figure]

**Figure 8.** Regional mean records of the last two century showing the recent warming period Red dashed lines correspond to linear trend obtained from Mann-Kendall test and black curve to ~50-years loess filtering. Dashed lines correspond to the 95% confidence level interval.

Commenté [NA15]: Confidence interval were add to judge the significance of the LOESS-filtering and the trend – Referee #2

[Figure]

[Figure]

**Figure 9.** Wavelet coherence analysis between subarctic North Atlantic mean record and instrumental AMO index during the 1856-2000 AD period. (a) Subarctic North Atlantic mean record (this study) and (b) instrumental AMO index (Enfield et al., 2001). Grey lines and dashed lines corresponds to filtered and wavelet reconstructed of the ~50-90 years and ~20-30 years periodicities highlighted on wavelet coherence spectrum (c). Colours on coherence wavelet spectrum represent correlation in both time and frequency domain. (Red equals highest correlation and blue lowest). White line corresponds to the cone of influence. Confidence level of 95% (α=0.05) is indicated with the black line.

[Figure]

**Figure 10.** Wavelet coherence analysis between Alaska mean record and instrumental PDO index during the 1900-2000 AD period. (a) Alaska mean record (this study) and (b) instrumental PDO index (Mantua et al., 1997). Grey lines corresponds to filtered and wavelet reconstructed of the ~16-30 years periodicity highlighted on wavelet coherence spectrum (c). Colours on coherence wavelet spectrum represent correlation in both time and frequency domain. (Red equals highest correlation and blue lowest). White line corresponds to the cone of influence. Confidence level of 95% ($\alpha$=0.05) is indicated with the black line.

[Figure]

**Figure 11.** Comparison between the North Atlantic mean record based on proxy data, the AMO index and the global Arctic sea-ice extent during their common period (1978-2000).

-----------------------------------------------Saut de page-----------------------------------------------
¶
**Figure 7.** Correspondence between the AMO index reconstructed multidecadal variability (grey line, Enfield et al., 2001) and the subarctic North Atlantic mean record (black line, this study) ¶
-----------------------------------------------Saut de page-----------------------------------------------
¶
*<objet>*¶
**Figure 8. Wavelet coherence analysis between Alaska mean record and instrumental PDO index during the 1900-2000 AD period. (a) Alaska mean record (this study) and (b) instrumental PDO index (Mantua et al., 1997). Grey lines corresponds to filtered and wavelet reconstructed of the ~16-30 years periodicity highlighted on wavelet coherence spectrum (c). Colours on coherence wavelet spectrum represent correlation in both time and frequency domain. (Red equals highest correlation and blue lowest). White line corresponds to the cone of influence. Confidence level of 95% (α=0.05) is indicated with the black line.**

---

## Author Response (AR2)

The authors and I would like to thank Reviewer #1 for their comments on our submitted manuscript.

Here, I would like to respond to and provide additional details on the Reviewer comments concerning the grouping into three regions (**comments #1, #2 and #3**).

Since the publication of the Arctic 2k database (McKay and Kaufman, 2014), reconstructions of climate variability obtained from the database were published either at the global arctic scale either for a specific region (Scandinavia) (Linderholm *et al.*, 2015). This both spatial scale do not allowed to take into account the role of climatic processes on arctic climate that are well known to have regional climatic impacts nowadays (*e.g.* climatic oscillations as AMO or PDO). We choose a regional approach to refine the comprehension of climate variability in the Arctic area and the three group were determined based on the regional impact of climatic oscillation found in the literature.

We agree that using an EOF analysis would be a good way to find the major patterns existing into the database. However, one of the main objectives of the paper is to determine the ability of the Arctic 2k database series to reproduce climate variability recorded in the observations data, especially during their common period. So that's why we choose to determine our regions based on the regional effect of internal atmosphere/ocean oscillations on climate that are currently recorded in instrumental data and not based on variability recorded in the palaeoclimate series. Regional effect of internal atmosphere/ocean oscillations on climate are describe in the manuscript (p4. L2-13).

Reviewer #1 also suggest to develop the part concerning the internal climate variability (**comment #4**). In fact, one of the most important result of our study is to highlight of the variabilities occurring at multidecadal scales record in paleoclimate data and linked to regional internal climate variability observed in instrumental data. We agree that the use of the recent observations allows us to determine the pattern of influence of PDO and AMO on climate in our study area but also including the role of the sea-ice cover variability. Using recent observations also allow us to compare it with our three regional mean records in order to determine the ability of them to reproduce the climate variability observed.

**References**

Linderholm H.W., Björklund J., Seftigen K., Gunnarson B.E. and Fuentes M.: Fennoscandia revisited: a spatially improved tree-ring reconstruction of summer temperatures for the last 900 years, Climate Dynamics , 45(3), 933-947, doi:10.1007/s00382-014-2328-9 2015.

McKay, N.P. and Kaufman, D.S.: An extended Arctic proxy temperature database for the past 2,000 years, Sci. Data 1:140026, doi: 10.1038/sdata.2014.26, 2014.

My co-authors and I would like to thank the anonymous Reviewer #2 for their comments and suggestions on our submitted manuscript. Here, I would like to provide addition details on the Reviewer #2 comments concerning a better rigorous treatment of uncertainty to put the trends and claimed covariabilities between our regional mean records and the instrumental climate index. For a better understanding, language comments are dissociated for the rest of the specific comments.

**Specific comments:**

- One of the comments of the Reviewer #2 is to be clear when **climatic means temperatures**". In the update version of the manuscript "Climatic" was modified where it's appropriate.

- The Reviewer #2 reports an arbitrary **regional grouping of the series**. The spatial density of the data set was the first argument to group the series into three regions, but the regional impact of climatic oscillation observed in instrumental data and found the literature allows to justify that this grouping has a currently climatic reality. In fact, one of the main objectives of the paper is to determine the ability of the Arctic 2k database series to reproduce the regional internal variability recorded in the observations data. May be this objective is not clearly notice in the introduction of the paper and we have to insist on that.

- p4L23**: Standardization** does not change the underlying distribution. It only changes the units. It is used here to compare the proxy series with different units. All the series were individually standardized before calculate each regional mean records.

- The **alpha (α) significance level** is the probability of rejecting the null hypothesis - data are independent and randomly ordered – and a significant trend exist. The term 'specific' is not really appropriate and it will be removed in the updated version of the manuscript.

- Concerning the **software packages used**, we propose to add the following sentence into the 2.4 Wavelet analysis section is add in the update version of the manuscript: 'Wavelet

analysis were performed with the software R (Team, 2008) using the packages biwavelet (Gouhier et al., 2012).'

- The size of the **axes labels on Fig. 2** are increase for a better readability.

- The Reviewer #2 suggest to **be more specific to explain the difficulty to identify the Little Ice Age** in the Arctic. A paragraph describing the different expressions of the LIA in Arctic will be added to the updated version of the manuscript. A new paragraph will also be added and the same synthesis will be made for the warm period of the MCA, with the update of the Fig. 4.

- It is notice that without uncertainties around regional mean records presented in Fig. 3 and Fig. 5, it is impossible to judge the **significance of the trends**. In the updated version of the manuscript Fig. 3 and Fig. 5, but also Fig. 6, will be modified and the standard deviation curves for the three regional mean records will be added. The addition of the number of records used for each regional mean records and the standard deviation is sufficient to evaluate the uncertainty around regional mean records and so the significance of the trends.

- p9 and 10: Indeed, **p-value** of 0.05 is commonly used for statistical test. If trend detected in the regional mean records are significant at 99% confidence level ($p < 0.01$), it is also true for a 95% confidence level ($p < 0.05$). Given that the 95% confidence level is used for the wavelet analysis, we will change this value to homogenize the new version of the manuscript. Autocorrelation and partial autocorrelation was calculated for the three regional mean records (see figures below). Results show that Mann-Kendall trend detected are not linked to autocorrelation in the regional mean records generated.

[Figure]

[Figure]

[Figure]

- Concerning the **uncertainty estimation for the wavelet analysis**, it is includes in the statistical test associated. For all local wavelet spectra, the statistical significance of peaks is assessed using Monte Carlo simulation against an appropriate background noise. Autoregressive modelling is used to determine the AR(1) stochastic process for each time series. AR(1) background noise could be either a red noise (AR(1)>0) or a white noise (AR(1)=0). Each AR(1) is calculated before performed wavelet analysis to

determine the background noir used. Concerning the cross-wavelet spectrum, detected fluctuations are statistically tested at α = 0.05 significance level against a red noise background. All these precision will be added in the 2.4. Wavelet Analysis part of the manuscript.

- In the update version of the manuscript the part about the link between climatic oscillation (AMO and PDO) and the proxy data will be developed, including the **role of sea-ice cover** on climate. Especially, a new figure will be added and presents the similarity between the trends of the AMO index and the sea-ice cover.

**Language comments:**

All the language mistakes listed by the anonymous Reviewer #2 were taken into account. Specifically:

- p1L18: replaced 'on many sort of proxy data' with 'on multiple proxy type records'
- p1L24: replaced 'show relationship' with 'show a relationship'
- p1L31: replaced 'temperature have' with 'temperature has'
- p2L5: added 's' to 'lake sediments'
- p2L7: added 's' to 'temperatures'
- p2L16: replaced the sentence 'The LIA is however characterized by an important spatial and temporal variability expression, particularly visible at more regional scale (e.g. Pages 2k Consortium, 2013).' with 'The LIA is known to have an important spatial and temporal variability, particularly at regional scale (e.g. Pages 2k Consortium, 2013).'
- p2L26: replaced 'led to' with 'were made to'
- p2L32: added 'a' to 'over a large spatial scale'
- p3L2: replaced 'data sets' with 'data set'
- p3L2: removed 'a' to 'A special attention'
- p4L21: replaced 'regional' with 'regionally'
- p5L5: removed 's' to 'trends'
- p5L8: replaced 'with' with 'which'
- p6L1: removed 's' to 'indicates'
- p6L6: replaced 'well-conservation' with 'conservation'
- p6L14: added 's' to 'decomposes'

- p8L3: replaced 'pronounced decreasing trend of temperatures' with 'decreasing temperatures'
- p9L8: removed 'date'
- p9L11: replaced 'seems' with 'appear'
- p9L14: 'removed 'the'
- p9L28: replaced 'to' with 'for'
- p9L29: added 's' to 'regions'
- The final paragraph was entirely corrected.

I would like to thank the anonymous Reviewer #3 for their comments and suggestions on the submitted manuscript. Here, I would like to response and provide additional details on the Reviewer #3 comments.

The anonymous Reviewer #3 note that the description of the Mann-Kendall, LOESS filtering and wavelet analysis is not necessary in the manuscript. To reduce the "**Methods" part**, we propose to include it part in several Appendix at the end of the manuscript.

In the new version of the manuscript, the **chapter "Secular variability"** will be modified. The description of the cold period of the LIA will be develop with the addition of a paragraph on the different way of characterizing the LIA in the Arctic area. The same synthesis will be made for the warm period of the MCA, with the update of the Fig. 4. The aims of these part is not the definition of forcings that can cause these two major climatic period and we will focus on the description of the temporal and spatial variability expression of the two periods.

The most important result of our study is the highlighting of variabilities occurring at multidecadal scales record in paleoclimate data and linked to regional internal climate variability observed in instrumental data. So in the update version of the manuscript the part about the link between climatic oscillation (AMO and PDO) and the proxy data will be developed. Especially, a new figure will be added and presents the similarity between the trends of the AMO index and the sea-ice cover to describe the interaction between internal climate variability, sea-ice cover fluctuations and climate variability record in our regional mean records. Because one of the main objectives of the paper is to determine the ability of the Arctic 2k database series to reproduce climate variability recorded in the observations data, we do not used the non-instrumental AMO and PDO records to go father back in time.

My co-authors and I would like to thank Dr. D.S. Kaufman for his comments on our submitted manuscript. Here, I would like to response to the essential additions for this paper highlighted by Dr. D.S. Kaufman:

**(1) Add a "Data Availability" section** :

In the updated version of the manuscript, a "Data availability" section is added with the Data URL for the Arctic 2k database v1.1.1 used in this study.

As suggested also by the Referee #2, the R software (Team, 2008) used to performed the wavelet analysis and the reference associated to the 'biwavelet' package (Gouhier et al., 2012) used for wavelet analysis will also be added in "2.4. Wavelet Analysis section" of the manuscript.

**(2) Add Data Citations and Publication Citations associated to the records** :

In the updated version of Supplementary Material associated to the manuscript, a new Table is added and replace Table S1. It will contains the description of proxy records in the Arctic 2k database arranged by the three regional regions used in the study, but also reference and it DOI for each record.

**(3) Submit the composite temperature time series by region to a public repository** :

We would like to clarify that we do not produce regional composite temperature time series by region but only regional mean records based on proxy data that has previously been standardized. The regional curves obtained will be published online after the publication of this article.

**(4) Why the analyses in this study are based on the old version of the PAGES 2k dataset?**

In the study, we used the Arctic 2k database v1.1.1 (Mc Kay and Kaufman, 2014) because it was the only version publicly available on the date of our manuscript submission (the February 28[th]). As you mentioned, a new Arctic 2k database exists but the reference paper associated is not still publicly available (PAGES 2k Consortium, *in press*; PAGES 2k Consortium, 2017). Without quality criteria of the database it will be difficult to estimate the influence of the use of a new version on the results. Moreover, due to the major difference between the two versions (19 records added and the suppression of 18 records), use the new version does not just correspond to an update of results but means redoing completely the study. This would be very interesting but requires more time and to be the topic of a new paper.

**(5) Archive a table that lists the beginning and ending of the LIA :**

A new Table S3 is added in the Supplementary Material. It will contains the starting and ending dates of the LIA, arranged by the three regional regions used in the study.

List of relevant changes made in the manuscript:

- **The introduction of the manuscript was modified** in order to better explain the objective of the paper: (1) study of the regional climate variability in the Arctic-subarctic area, (2) determine the ability of the proxy records to reproduce instrumental climate variability and (3) role of the internal climate variability during the last two centuries. **The conclusion was also rewrite**.

- To **reduce the** "**Methods" part**, the description of the Mann-Kendall, LOESS filtering and wavelet analysis was include in **several Appendix** at the end of the manuscript.

**- A new part presenting comparison between a global Arctic mean record and the three regional curves was added and developed** to add precision concerning the grouping into three groups. The choice of regional approach is now explain by the spatial distribution of the serie and the groupping is justified by actual regional climatic influence.

- **Confidence interval were add to judge the significance of the LOESS-filtering and the trend in Figures 5 and 8.**

- **The "Secular variability part" was developed** with a synthesis of the expression of the MCA in the Arctic-subarctic region and **a new figure was added** (Fig. 6). **Two supplementary Tables and Figures was also added** (beginning and ending of the LIA and the MCA)

- **A new figure was added and presents the similarity between the trends of the AMO index and the sea-ice cover** to describe the interaction between internal climate variability, sea-ice cover fluctuations and climate variability record in our regional mean records.

- The entire manuscript was corrected by a native English-speaking editor (Edanz, see order in attachment) which help scientists and researchers prepare their work for publication for 20+ years and have built a reputation as the leading author services provider.

Order #3758 was placed on 2017-08-09 and is currently Completed.

**ORDER DETAILS**

| PRODUCT | TOTAL |
|---|---|
| Native English Check [https://nec.edanzediting.com/product/native-english-check/] × 5650 | $339.00 |
| SUBTOTAL: | $339.00 (ex. tax) |
| ADD FILE ATTACHMENT: - UPLOADED FILE [HTTPS://NEC.EDANZEDITING.COM/WP-CONTENT/UPLOADS/2017/08/RELECTURE-598A486FD8BF0.DOCX] | $0.00 |
| PAYMENT METHOD: | PayPal |
| TOTAL: | $339.00 |

**DOWNLOAD EDITED FILES**

N1708-3758-Debret-Relecture-TrackedCopy-1.docx [https://nec.edanzediting.com/wp-content/uploads/2017/08/N1708-3758-Debret-Relecture-TrackedCopy-1.docx]

N1708-3758-Debret-Relecture-ClearCopy.docx [https://nec.edanzediting.com/wp-content/uploads/2017/08/N1708-3758-Debret-Relecture-ClearCopy.docx]

Order again [/my-account/view-order/3758/?order_again=3758&_wpnonce=b04c6ebc21]

**CUSTOMER DETAILS**

| EMAIL: | maxime.debret@univ-rouen.fr |
|--------|------------------------------|
| PHONE: | 0033689010810 |

**Billing address**

Université de Rouen
Maxime Debret
Place Emile Blondel
76821 MONT SAINT AIGNAN
France

Copyright © Edanz Group Ltd. 2000-2017

[revised manuscript text omitted]

Commenté [NA1]: Modified in order to better explain the objective of the paper: (1) study of the regional climate variability in the Arctic-subarctic area, (2) determine the ability of the proxy records to reproduce instrumental climate variability and (3) role of the internal climate variability during the last two centuries – Referee #1, Referee #2 and Referee #3

~~Over the last decade, extensive efforts led to collect and centralize palaeoclimate data available in order to reconstruct past climate variability at regional, hemispheric and global scales. Most temperature reconstructions include different types of archives and proxies (Morberg et al., 2005; Mann et al., 2009; Kaufman et al., 2009; Ljungqvist, 2010; Marcott et al., 2013) and some studies focused on single palaeoclimate archive type and/or area (e.g. McGregor et al., 2015 for oceans; Weissbach et al., 2016 for ice core; Wilson et al., 2016 for tree rings). Recently, the publication of high time resolution reconstructions by the PAGES 2k Consortium (PAGES 2k Consortium, 2013) and particularly for the Arctic area (Hanhijärvi et al., 2013; McKay and Kaufman, 2014), offers the possibility to study the spatial and temporal pattern of climate variability over large spatial scale from low frequencies (i.e. millennial and multi-centennial fluctuations) to high frequencies such as decadal variations.~~

~~In this paper, we use statistical and wavelet analysis in order to characterize long-term and secular (Little Ice Age, LIA) climatic fluctuations that occurred in the Arctic during the past 2000 years, based on a regional data sets. A special attention is given to the last two centuries with the aim to document the respective responses of the climate system to anthropogenic forcing and internal climate variability in the Arctic.~~

**2. Paleoclimate data**

The records used in this study were compiled by the Arctic 2k working group of the Past Global Changes (PAGES) research program. This working group released a database  comprising 56 proxy  records for the Arctic area (version 1.1.1, McKay and Kaufman, 2014).

The database contains all available records  that meet  data quality criteria concerning location (from north of 60°N), time coverage (extend back to at least 1500 A.D.), mean resolution ( better than 50 years), and dating control (at least one age control point every 500 years) (Fig. 1a). See Table S1 in supplementary material for more information about each site  (cf. also McKay and Kaufman  2014).

Proxy records are from different archive types. Most are continental archives with very reliable chronologies (16 ice cores, 13 tree rings, 19 lake sediment cores and 1 speleothem). Six records are from marine archives and one is a historic record (months of ice cover). Among the 56 records, 35 have an annual resolution  (Fig. 1b). Hence,  the high temporal  resolution of the  Arctic 2k database series offers the possibility to study the high frequency climate variability of the last two  millennia, assuming that the proxy record climate variability and the archiving process do not induce a bias in the multi-annual to centennial frequencies analyzed.

The database has been built from palaeoclimate proxy series with demonstrated relationship to temperature variability. All the proxy data used have been published in peer-reviewed journal and the sensitivity of each proxy record to temperature was

**Commenté [NA2]:** The analyses are based on the v1.1.1 PAGES 2k dataset because it was the only version publicly available on the date of the manuscript submission. Now, a new version is available but the major difference between the two versions requires a thorough study of the influence of the updating of the database and therefore a new study. – Comment (4) – D.s Kaufman

**Commenté [NA3]:** Table S1 was replace by a new table S1 which contains the description and the URL for each proxy records, arranged by the three regional regions – Comment (2) – D.s Kaufman

[revised manuscript text omitted]

**Commenté [NA11]:** Added to precise concerning the way of grouping the data into three groups. The regional approach is now explain by the spatial distribution of the serie and the groupping is confirmed by actual regional climatic influence - Comments (1), (2) and (3) - Referee#1

[Figure]

**Figure 4.** *Left.* Correlation between the global mean based on proxy data and the three regional mean records for (a) North Atlantic (b) Alaska and (c) Siberia areas. Correlations is significant at the 95% confidence level. *Right.* (e) Wavelet coherence between global and North Atlantic mean records, (f) global and Alaska mean records, (g) global and Siberia mean records. Colors represent the amplitude of the signal at given time and spectral period (red equals highest power, blue lowest). White line corresponds to cone of influence on wavelet coherence spectrum. Confidence level of 95% (α=0.05) is indicated on wavelet spectrum with the black line.

**Commenté [NA12]:** Developed to precise concerning the way of grouping the data into three groups. The regional approach is now explain by the spatial distribution of the serie and the groupping is confirmed by actual regional climatic influence - Comments (1), (2) and (3) - Referee#1

[Figure]

Figure 5. (a) Individual trends for each records before recent warming. White dot highlighted inconsistency between two tendencies for the same archive. North Atlantic (b), Alaska (c) and Siberia (d) regional 50-years LOESS. Blue colors indicate decreasing tendency whereas red colors indicate increasing trends. Dashed black lines correspond to the 95% confidence interval.

Commenté [NA13]: Confidence interval were add to judge the significance of the LOESS-filtering and the trend – Referee #2

[Figure]

**Figure 6.** Expression of the Medieval Climate Anomaly (MCA) of the Arctic 2k series based on references paper (see McKay and Kaufman, 2014): starting (a), ending (b) and length (c). Symbols in grey correspond to series for which the MCA is not mentioned by the authors. More details concerning the temporal expression of the LIA are available in Table S4 and Figure S2.

Commenté [NA14]: Added to also describe the expression of the warm MCA in the Arctic area – Referee #3

[Figure]

**Figure 47.** Spatial expression of the Little Ice Age (LIA) of the Arctic 2k series based on references paper (see McKay and Kaufman, 2014): starting (a), ending (b) and length (c). Symbols in grey correspond to series for which the LIA is not mentioned by the authors in the original publication. More details concerning the temporal expression of the LIA are available in Table S3 and Figure S1.

[Figure]

**Figure 5.** Regional mean records of the last two century showing the recent warming period Red dashed lines correspond to linear trend obtained from Mann-Kendall test and black curve to ~50-years loess filtering. Dashed lines correspond to the 95% confidence level interval.

Commenté [NA15]: Confidence interval were add to judge the significance of the LOESS-filtering and the trend – Referee #2

[Figure]

**Figure 6.9.** Wavelet coherence analysis between subarctic North Atlantic mean record and instrumental AMO index during the 1856-2000 AD period. (a) Subarctic North Atlantic mean record (this study) and (b) instrumental AMO index (Enfield et al., 2001). Grey lines and dashed lines corresponds to filtered and wavelet reconstructed of the ~50-90 years and ~20-30 years periodicities highlighted on wavelet coherence spectrum (c). Colours on coherence wavelet spectrum represent correlation in both time and frequency domain. (Red equals highest correlation and blue lowest). White line corresponds to the cone of influence. Confidence level of 95% (α=0.05) is indicated with the black line.

[Figure]

**Figure 810.** Wavelet coherence analysis between Alaska mean record and instrumental PDO index during the 1900-2000 AD period. (a) Alaska mean record (this study) and (b) instrumental PDO index (Mantua et al., 1997). Grey lines corresponds to filtered and wavelet reconstructed of the ~16-30 years periodicity highlighted on wavelet coherence spectrum (c). Colours on coherence wavelet spectrum represent correlation in both time and frequency domain. (Red equals highest correlation and blue lowest). White line corresponds to the cone of influence. Confidence level of 95% ($\alpha=0.05$) is indicated with the black line.

[Figure]

**Figure 11.** Comparison between the North Atlantic mean record based on proxy data, the AMO index and the global Arctic sea-ice extent during their common period (1978-2000)**.**

[Figure]

**Figure 7.**

[Figure]

**Figure 8. Wavelet coherence analysis between Alaska mean record and instrumental PDO index during the 1900-2000 AD period. (a) Alaska mean record (this study) and (b) instrumental PDO index (Mantua et al., 1997). Grey lines corresponds to filtered and wavelet reconstructed of the ~16-30 years periodicity highlighted on wavelet coherence spectrum (c). Colours on coherence wavelet spectrum represent correlation in both time and frequency domain. (Red equals highest correlation and blue lowest). White line corresponds to the cone of influence. Confidence level of 95% (α=0.05) is indicated with the black line.**